# Modelling soil and landscape evolution– the effect of rainfall and land use change on soil and landscape patterns

W. Marijn van der Meij[1,2], Arnaud J. A. M. Temme[3,4], Jakob Wallinga[1], Michael Sommer[2,5]

[1]Soil Geography and Landscape Group, Wageningen University and Research, P.O. Box 47, 6700 AA, Wageningen, The Netherlands
[2]Research Area Landscape Functioning, Working Group Landscape Pedology, Leibniz-Centre for Agricultural Landscape Research ZALF, Eberswalder Straße 84, 15374 Müncheberg, Germany
[3]Department of Geography, Kansas State University, 920 N17th Street, Manhattan, KS 66506, USA
[4]Institute of Arctic and Alpine Research, University of Colorado, Campus Box 450, Boulder, CO, 80309-0450, USA
[5]Institute of Environmental Science & Geography, University of Potsdam, Karl-Liebknecht-Straße 24-25, 14476 Potsdam, Germany

Correspondence to: W. Marijn van der Meij (marijn.vandermeij@wur.nl)

**Abstract.** Humans have substantially altered soil and landscape patterns and properties due to agricultural use, with severe impacts on biodiversity, carbon sequestration and food security. These impacts are difficult to quantify, because we lack data on long-term changes in soils in natural and agricultural settings and available simulation methods are not suitable to reliably predict future development of soils under projected changes in climate and land management. To help overcome these challenges, we developed the HydroLorica soil-landscape evolution model, that simulates soil development by explicitly modelling the spatial water balance as driver of soil and landscape forming processes. We simulated 14500 years of soil - formation under natural conditions for three scenarios of different rainfall inputs. For each scenario we added a 500-year period of intensive agricultural land use, where we introduced tillage erosion and changed vegetation type.

Our results show substantial differences between natural soil patterns under different rainfall input. With higher rainfall, soil patterns become more heterogeneous due to increased tree throw and water erosion. Agricultural patterns differ substantially from the natural patterns, with higher variation of soil properties over larger distances and larger correlations with terrain position. In the natural system, rainfall is the dominant factor influencing soil variation, while for agricultural soil patterns landform explains most of the variation simulated. The cultivation of soils thus changed the dominant factors and processes influencing soil formation, and thereby also increased predictability of soil patterns. Our study highlights the potential of soil-landscape evolution modelling for simulating past and future developments of soil and landscape patterns. Our results confirm that humans have become the dominant soil forming factor in agricultural landscapes.

## 1    Introduction

Soils provide valuable functions for nature and society by supporting plant growth and agriculture, managing water and solute flow, sequestering carbon, preserving archaeological heritage, creating habitats for plants and animals and providing support for infrastructure (Dominati et al., 2010; Greiner et al., 2017). However, soils are currently degrading by agricultural intensification and climate change, forming one of the largest threats to global food security and biodiversity (Bai et al., 2008; Montanarella et al., 2016; Tscharntke et al., 2012). A drastic change in land management is needed to restore healthy soils and soil functions (IPCC, 2019). Combating soil degradation and promoting sustainable land management therefore stands high on the agenda of the soil science community (Bouma, 2014; Cowie et al., 2018; Keesstra et al., 2018; Kust et al., 2017; Minasny et al., 2017).

The first step towards sustainable land management and a return to healthy, natural soils is a fundamental understanding of the development and characteristics of natural soil patterns, and how these change under human influence. Therefore, we will focus in this paper on gently to strongly sloping undulating landscapes that are suitable for agricultural use (max slope ~20%, Bibby and Mackney, 1969). Soil forming processes are controlled by at least five environmental factors: climate, organisms, relief, parent material and time (the ClORPT model, Jenny, 1941). Different factors dominate in natural and agricultural settings. In natural settings with flat or undulating topography, soil erosion generally occurs at very low rates or is absent (Alewell et al., 2015; Wilkinson, 2005). Some soil redistribution can occur as a consequence of creep or tree throw (Gabet et al., 2003). More importantly, tree throw creates local pits and mounds, which temporarily change hillslope hydrology and act as local hotspots for soil development due to a larger influx of water (Šamonil et al., 2015; Shouse and Phillips, 2016). These seemingly random processes create a high degree of heterogeneity in soil patterns, which shows little to no correlation with relief (Vanwalleghem et al., 2010). In contrast, intensively managed agricultural landscapes show soil patterns that closely follow the relief (Phillips et al., 1999; Van der Meij et al., 2017). This reflects that erosion processes are relief-dependent and this propagates into the soil patterns, unless erosion and deposition patterns are affected by field margins such as hedges or banks. The switch from such natural to agricultural soil systems can occur abruptly, e.g. by deforestation or the implementation of highly mechanized agriculture in a few decades. Sommer et al. (2008) described this switch in boundary conditions and its implications with a time-split approach: Over a short time period – relative to Holocene soil evolution - the soil system changes from natural, progressive pedogenesis, where profile deepening and horizon formation dominate erosive processes, to regressive pedogenesis, where - vice versa - erosion and deposition dominates progressive pedogenic processes (Johnson and Watson-Stegner, 1987).

The coexistence of both progressive and regressive processes in a defined period of time has been described by several authors. In a progressive phase there are also regressive processes that change soils, terrain and hydrological pathways (Phillips et al., 2017; Šamonil et al., 2018). In a regressive phase, progressive processes still have a substantial effect on soil development (Doetterl et al., 2016; Montagne et al., 2008). Colluvic soils might be influenced by groundwater or subject to continuous clay illuviation (Leopold and Völkel, 2007; Van der Meij et al., 2019, SI; Zádorová and Penížek, 2018). Furthermore, the changes

in boundary conditions are not always abrupt as e.g. deforestation. Historic erosion processes with rates much lower than current erosion processes might have given pedogenic processes the time to alter soil and colluvium (Van der Meij et al., 2019).

To disentangle complex history and causes of soil formation, data is required on both natural and agricultural soils that have formed under similar conditions, and preferably from the same region. However, there is limited undisturbed natural land left, often rapidly declining, in places that are unsuitable for agriculture, and/or indirectly influenced by anthropogenic climate change (e.g. tropical and boreal zones, IPCC, 2019). Moreover, (historical) cultivation occurred in areas and soils most suitable for agriculture (Pongratz et al., 2008; Vanwalleghem et al., 2017), leaving less suitable land undisturbed. This complicates comparison and empirical inference. Because of the complex interactions between pedogenic and geomorphic processes, and the lack of field data, we heavily depend on process knowledge and model simulations for mechanistic inference about how natural soil patterns develop as function of their environments and how this changes in agricultural settings (Opolot et al., 2015).

Soil evolution models simulate a range of physical, chemical and biotic processes that affect the properties of soils through space and time (Minasny et al., 2015; Stockmann et al., 2018; Vereecken et al., 2016). Such models have been developed for a range of scales, varying from 1D soil profiles to 3D soil landscapes (Finke, 2012; Minasny et al., 2015; Temme and Vanwalleghem, 2016). One-dimensional soil profile models generally provide a high level of detail and process coverage, but they lack the simulation of essential feedbacks and interactions that can occur between soils on a landscape scale (Van der Meij et al., 2018). For example, the spatial redistribution of water or the exchange of soil material through erosion and deposition processes affect soils differently at different landscape positions. Soil landscape evolution models (SLEMs) do simulate lateral distribution of solids by geomorphic processes and consider soils as continua rather than discrete units. Current SLEMs perform reasonably well in landscapes where lateral soil movement is substantial (e.g. Temme and Vanwalleghem, 2016; Van Oost et al., 2005). However, these models are not developed to simulate soil development in relatively stable landscapes where lateral water redistribution is the dominant driver causing soil heterogeneity, because this hydrologic control is not explicitly modeled (Van der Meij et al., 2018).

To summarize, we are currently lacking data and methods that can quantify the effect of changing soil forming factors on soil development and spatiotemporal soil patterns. This knowledge is essential for the transition to sustainable land management and adaptation to the changing climate. The objective of this study is to develop a suitable model to quantify the variation and predictability of soil patterns as a function of varying environmental factors. We will address three questions:

1. What are the basic characteristics of soil patterns in natural and agricultural landscapes?
2. What are the major factors driving soil formation in natural and agricultural landscapes?
3. How does the predictability of soil patterns change through time and after cultivation?

We developed a soil-landscape evolution model that can simulate natural soil and landscape evolution by incorporating dominant natural processes such as soil creep, tree throw, vegetation dynamics and infiltration-dependent pedogenesis driven

by the soil forming factors climate, organisms, relief, parent material and time. We simulated soil formation for 14500 years under three scenarios of rainfall (dry, humid, wet) to quantify the effect of water availability and distribution on soil variation in natural systems. Each run was concluded with 500 years of intensive agricultural land use, where we introduced the process of tillage erosion. Tillage erosion is a dominant process redistributing soil material in intensively managed agricultural fields (Van Oost et al., 2005).

We expect that before intensive cultivation, spatial soil heterogeneity will be larger for greater rainfall, due to more intense erosion and translocation processes, and effects of vegetation. Moreover, we expect that the spatial heterogeneity increases by erosion processes under cultivation, also resulting in larger correlations between soil properties and topographic properties, because of the topographic dependence of erosion processes. This would imply that soil patterns become more predictable due to cultivation, at least for circumstances without hedges or banks that would modify the spatial distribution of erosion and deposition areas.

For our simulations, we created a hypothetical loess-covered, hilly landscape with a range of characteristic slope positions as spatial setting. We choose loess, because it is a relatively homogeneous parent material, widely spread globally and favored for agricultural practices due to its high water holding capacity and resulting fertility (Catt, 2001). The long-term use of loess areas for agriculture and unsustainable management has resulted in severe land degradation (e.g. Zhao et al., 2013).

## 2    Methods

### 2.1    Model

Here we describe our model named HydroLorica. HydroLorica is based on the model Lorica (Temme and Vanwalleghem, 2016), but includes explicit simulation of water flow and water availability as drivers of natural soil, landscape and vegetation change (Van der Meij et al., 2018). HydroLorica is a reduced-complexity model, which means that it simulates processes affecting soil and landscapes using simplified process descriptions. Reducing model complexity promotes critical evaluation of essential processes, reduces calculation time and prevents extensive data requirements and over-parameterization (Hunter et al., 2007; Kirkby, 2018; Marschmann et al., 2019; Snowden et al., 2017; Temme et al., 2011).

#### 2.1.1    Model architecture

HydroLorica is a raster-based model, where a Digital Elevation Model (DEM) determines the shape of the terrain. Below each raster cell of the DEM there is a predetermined number of soil layers with layer thicknesses variable in space and time. Each layer can contain a specific mixture of gravel, sand, silt and clay and two types of organic matter (quickly and slowly decomposing, Yoo et al., 2006), depending on parent material and occurring pedogenic processes. Pedogenic and geomorphic processes affect the contents of the layers, leading to differences in soils in space and time. Changes in soil properties and contents modify layer thicknesses and surface elevation through a pedotransfer function (PTF) of bulk density. The use of a pedotransfer function allowed the model to calculate variations in layer thicknesses due to pedogenic and geomorphic processes. We used the same PTF for bulk density as the original Lorica model (Tranter et al., 2007). We refer to Temme and Vanwalleghem (2016) for more information about the spatial model architecture of Lorica, which we maintained in our adaptation HydroLorica. In this project, we worked with 25 soil layers, with an initial uniform thickness of 0.15 m. When a layer got very thick or very thin (55% thicker or thinner than its initial value), the layer was split or combined with another layer.

The annual changes in texture classes *tex* [kg] and organic matter classes *om* [kg] in layer *l* at location *xy* and time *t* are governed following Eqs. (1) and (2) (for abbreviations of processes, see Table 1). The changes in mass of texture and organic matter are converted to a change in layer thickness [m] using a pedotransfer function (Tranter et al., 2007). We calculated the bulk density of the fine mineral fraction [kg m$^{-3}$] with Eq. (3) using the sand and silt fraction [-] and the depth below the surface [m]. HydroLorica includes a correction of bulk density taking into account the effects of the coarse fraction and the organic fraction using Eq. (4), using a density of 2700 kg m$^{-3}$ for the coarse fraction (Temme and Vanwalleghem, 2016) and a density of 224 kg m$^{-3}$ for the organic fraction (Tranter et al., 2007). In our study, there is no coarse soil material present. This pedotransfer function does not directly take into account changes in bulk density stemming from soil structuring, weathering or bioturbation. Instead, depth below the surface is used as proxy for these factors. The used PTF has a relatively low fit with the data it was derived from ($R^2 = 0.41$, Tranter et al., 2007). However, PTFs that yield a higher accuracy often require advanced calculation methods (Chen et al., 2018; Ramcharan et al., 2017) or soil properties that are not readily available in HydroLorica.

As we discuss in Van der Meij et al. (2018), the estimation of such properties often gives biased or highly uncertain results, which would propagate into the calculation of bulk density. Rather than stacking pedotransfer functions, we decided to use a PTF that required input that is readily available in HydroLorica and could be calculated within the model itself.

The sum of changes in layer thickness of all layers $L$ calculated through changes in bulk density and mass of the layers result in the annual change of elevation $z$ (Eq. (5)). Clay translocation and water erosion are directly driven by the total annual water flow, while occurrence of tree throw and rates of creep, bioturbation and organic matter accumulation are indirectly driven by water availability via vegetation controls. Infiltration $I$ is the difference between precipitation $P$ and spatially explicit actual evapotranspiration $ETa$, runon $ROnn$ and runoff $ROff$ (Eq.(6)). HydroLorica works with dynamic time steps as suggested by

Van der Meij et al. (2018) to capture process dynamics at their relevant scales, while optimizing calculation time. Hydrologic processes are calculated with a daily, monthly, or yearly time step, with smaller timesteps selected during wetter conditions for more accurate simulation. Annual sums of infiltration and overland flow are used to drive geomorphic, pedogenic and biotic processes.

$$\Delta tex_{xy,l,t} = \Delta tex_{CR,xy,l,t} + \Delta tex_{WE,xy,l,t} + \Delta tex_{TT,xy,l,t} + \Delta tex_{TI,xy,l,t} + \Delta tex_{CT,xy,l,t} + \Delta tex_{BT,xy,l,t} \tag{1}$$

$$\Delta om_{xy,l,t} = \Delta om_{CR,xy,l,t} + \Delta om_{WE,xy,l,t} + \Delta om_{TT,xy,l,t} + \Delta om_{TI,xy,l,t} + \Delta om_{CAB,xy,l,t} + \Delta om_{BT,xy,l,t} \tag{2}$$

$$BD_{fine,xy,l,t} = 1000(1.35 + 0.452(f_{sand} + 0.76f_{silt}) + (100(f_{sand} + 0.76f_{silt}) - 44.65)^2 * -0.000614 + 0.06 * \log_{10}(depth)) \tag{3}$$

$$BD_{soil,xy,l,t} = \frac{mass_{total,xy,l,t}}{\frac{mass_{fine,xy,l,t}}{BD_{fine,xy,l,t}} + \frac{mass_{coarse,xy,l,t}}{2700} + \frac{mass_{organic,xy,l,t}}{224}} \tag{4}$$

$$\Delta z_{xy,t} = \sum_{l=1}^{L} \Delta BD_{soil}\left(\sum tex_{xy,l,t} + \sum om_{xy,l,t}\right) \tag{5}$$

$$I_{xy,t} = P_t - ETa_{xy,t} + ROnn_{xy,t} - ROff_{xy,t} \tag{6}$$

### 2.1.2 Process formulation and parameters

In our model we considered only the impact of physical and biological processes on soil properties. The current model architecture does not facilitate the simulation of soil chemical processes. The selected processes are described below. Drivers and impacts of each process are summarized in Table 1. We summarized the drivers per soil forming factor. We mostly used the processes and parameters of Lorica as reported in Temme and Vanwalleghem (2016), which we summarize here. When we added a new process, or changed its parameters, the adjustments are reported in this Section. We provided a detailed

overview of the equations and selected parameters in Supplement 1.

We aim to understand the functioning of general soil landscape systems. Therefore, we parametrized and calibrated the model processes using regional data or process rates from literature that are valid for larger regions. We did not calibrate the parameters on data from one specific study site to avoid the effect of any idiosyncrasies that can be present in that data. For other processes where there was no regional data available, we estimated the parameters so that the effects of those processes

were in the same order of magnitude as processes with rates based on literature. An overview of the process parameters is provided in table S1.

#### 2.1.2.1  Hydrologic processes

The hydrological module partitions spatially uniform rainfall (P) into three spatially explicit components: evapotranspiration (ET), infiltration (I) and surface flow (Ronn & ROff, Eq. (6)). Potential ET is calculated from prescribed temperature using the Hargreaves-Samani equation (Hargreaves and Samani, 1985), and corrected for topographical position (Swift Jr, 1976) and vegetation type (Allen et al., 1998). Surface flow is calculated on a daily basis, and only when rainfall intensity [amount / duration, mm hr$^{-1}$] exceeds the saturated hydraulic conductivity of the topsoil, which is a function of soil properties and slope (Morbidelli et al., 2018; Wösten et al., 2001), or precipitation in the form of snow is melting. The excess water is routed over the surface using the multiple flow algorithm (Holmgren, 1994) and can re-infiltrate in places with higher hydraulic conductivity, in local surface depressions, or can leave the catchment. HydroLorica can thus deal with DEMs that contain depressions, and actively forms depression by simulating tree throw. The annual sum of daily surface flow is used to calculate annual water erosion and deposition using the stream power law. To account for seasonal differences, actual ET is calculated on a monthly basis from the potential ET and rainfall using the topsoil water budget model of Pistocchi et al. (2008). Infiltration is the sum of (re-)infiltrated surface water and the monthly difference between rainfall and actual ET (Eq. (6)). The annual water balance is used as a driver of various geomorphic and pedogenic processes, and to determine vegetation type. The hydrological module is described in detail in Appendix A of Van der Meij et al. (2018).

#### 2.1.2.2  Determination of vegetation type

We considered two types of natural vegetation: grassland and forest. The vegetation type depends on the water availability; where rainfall plus re-infiltration exceeds potential evapotranspiration, there is no water stress and forests can grow. Otherwise, there is water stress and there will be grassland. This threshold is based on a hypothesis from Thompson et al. (2010), who used the Budyko curve (Budyko and Miller, 1974) to estimate vegetation type. By extending this relationship with re-infiltration, this relation can be used to assess local, but spatially explicit vegetation type. Vegetation type thus has a climatic control and a topographic control in the form of hillslope aspect and local convergence of water flow in gullies and depressions (e.g. Metzen et al., 2019). This variation in moisture and vegetation can occur very locally, especially in semi-arid regions. Vegetation type influences evapotranspiration (Allen et al., 1998), bioturbation and creep rate (Gabet et al., 2003), the occurrence of tree throw, and also controls organic matter input. Under intensive agricultural use, we convert the vegetation type to arable crops. We assume that soil and landscape processes are similar to landscapes under grassland vegetation. The differences are that arable crops have lower potential evapotranspiration and the process of tillage is introduced.

Our method of estimating vegetation type can lead to annual changes in vegetation type depending on water availability, because we do not consider ecological processes such as resilience or succession. The portion of years with grassland and forest vegetation aggregated over longer time spans (> 100 a) provides an estimate for the forest cover of that specific location (see the animations in Supplement 2). The vegetation distribution should thus be considered on an aggregated level rather than an annual level to yield meaningful results. This implementation suffices for our focus on long-term changes in soils and terrain, but should not be used to study systems on annual to decadal time scales.

### 2.1.2.3 (Bio-)geomorphic processes

The main (bio-)geomorphic processes affecting topography in loess areas are soil creep, tree throw, water erosion and tillage erosion. Soil creep is a bio-geomorphic process that causes a diffuse movement of soil material on a hillslope, driven by various factors such as (micro)climate, organisms and terrain (Pawlik and Šamonil, 2018; Regmi et al., 2019; Roering et al., 2002). The potential creep rate is a function of vegetation type and slope (Gabet et al., 2003). We adopt higher creep rates in forested areas, because of the deeper rooting depth and higher root abundance. We divided the potential creep rate at a certain location over all soil layers, with exponentially decreasing rates deeper in the soil. The transport of soil material from a layer to layers in its lower lying neighboring cells is proportional to the surface slope and shared layer boundaries.

Tree throw is a bio-geomorphic process that has a distinct effect on the terrain and water routing; the created pit can act as hotspot for soil formation by the increased infiltration of water (Šamonil et al., 2018). We simulated tree throw as a random process, with on average 0.2 trees falling per hectare per year. This rate is lower than other rates found in natural forests around the world (0.3-1.5 trees ha-1 a-1, Finke et al., 2013; Gallaway et al., 2009; Phillips et al., 2017), because some factors controlling tree uprooting like shallow rooting depths due to impermeable layers or steep slopes are not present in our spatial setting. The dimensions of the root clump that is transported by tree throw were scaled with the age of the falling tree, which was also randomly selected. We assumed that tree growth occurs in the first 150 years of a tree's existence, after which size remains stable until a maximum age of 300 years. These numbers and trends are loosely based on Rozas (2003). A pit and mound topography is only formed when the dimensions of the root clump exceed the size of the raster cell (1.5 m in our case) and that material is transported to a cell downslope. When the root clump is smaller than the cell size, or when the slope of the terrain does not lead to downward transport of the material, tree throw will only cause a (partial) turbation of the upper layers in the affected raster cells.

Water erosion and deposition are calculated using the same approach as the original Lorica model (Temme and Vanwalleghem, 2016). Sediment uptake and deposition are calculated as function of discharge and surface gradients (Schoorl et al., 2002). Sediment uptake is simulated as a selective process, where smaller particles are easier to erode and more difficult to deposit. Organic matter behaves the same as clay under erosion, because we assumed that organic matter occurs in associations with clay particles. Water erosion is limited by the occurrence of coarse soil particles (surface armoring) and vegetation. The role of water erosion in forested loess catchments is limited (Vanwalleghem et al., 2010); the vegetation protects the soil below from erosion. However, disturbances such as forest fires can temporarily increase erodibility of the soil. Therefore, we did simulate water erosion in forested landscapes, but with lower rates than in grassland. We simulated this by including a high vegetation protection constant (value of 1) in forested sites. In grasslands we used the aridity index between 0 and 1 as vegetation protection constant.

Tillage erosion was simulated as a diffusive process, similar to creep, with some differences: tillage homogenized the soil over the reach of the plough depth, erosion only occurred from the top layer contrary to the whole soil profile as with creep, and the erosion rates were much higher due to the intensive land management.

### 2.1.2.4 (Bio-)pedogenic processes

We simulated three dominant (bio-)pedogenic processes that change texture and organic matter properties in loess landscapes. These are clay translocation, bioturbation and soil organic matter accumulation and breakdown.

We adapted a new way of simulating clay translocation, using the advection equation of Jagercikova et al. (2017). The diffusive part of clay translocation as described by Jagercikova et al. (2017) is separately modeled by bioturbation. We scaled the parameters of clay translocation with local infiltration to develop an infiltration-dependent equation. Not all clay in the soil is

245 available for translocation. Part of it is not available to the percolating water, because it is bonded to other minerals and organic matter. We used the equations of Brubaker et al. (1992) to estimate the part of the clay that is water-dispersible, i.e. that is available for translocation by water. We estimated the required CEC with a pedotransfer function from Ellis and Foth (1996), as a function of clay content and organic matter content. Following from these equations, the fraction of non-dispersible (remaining) clay is 5.9% in soils without SOM and increases with 1.2% for every extra percent of SOM. This approach is

250 similar to the one used in soil profile model SoilGen2 (Finke, 2012).

Bioturbation works as a diffusive processes, homogenizing the soil vertically (Yoo et al., 2011). We used the same rates for bioturbation as for creep, because these processes are driven by the same organisms reworking the soil. The potential bioturbation rate was divided over each soil layer by integrating the exponential depth function over the layer thickness, and then dividing by the integration of the function over the entire soil profile. Every layer exchanges a certain fraction of its

contents, based on initial bioturbation rate and depth, with all other layers. The amount of exchange between two layers decreases with increasing distance.

Soil organic matter (SOM) accumulation and breakdown was simulated as in earlier soil-landscape evolution models (Minasny et al., 2008; Temme and Vanwalleghem, 2016; Vanwalleghem et al., 2013; Yoo et al., 2006). Accumulation of SOM is controlled by the potential input and depth in the soil. The accumulation is divided over a young and old SOM pool using a

260 fractionation factor. These pools differ in their rate of decomposition. We calibrated the SOM cycle in agricultural settings with the average depth distribution of organic carbon in agricultural soils on the Chinese loess plateaus (Liu et al., 2011). We simulated 5000 years of soil development using different process parameters. We selected the parameter set that simulated an organic matter distribution most similar to the reference distributions from Liu et al. (2011). The reported depth distributions for pasture and forest soils by Liu et al. (2011) were not useful for this project. Soils under these vegetation types on the

265 Chinese loess plateau generally contain lower SOM stocks than natural landscapes, because these positions often have recently been replanted to combat soil erosion or because they occur on topographic positions which are not favorable for plant growth and agriculture. Instead, we calculated reference carbon stocks for forest and grassland soils by adjusting the agricultural carbon stocks of Liu et al. (2011) with changes in carbon stocks after conversion from forest to crop and from forest to pasture (Guo and Gifford, 2002). With the resulting reference carbon stocks for natural vegetation we ran additional calibrations to

270 calculate the potential SOM input for forest and grassland.

## 2.2 Experimental setup

We developed an artificial topographic setting in which we performed our simulations. The use of an artificial setting rather than a field setting avoids the effect of local disturbances and idiosyncrasies which can disturb general signals we look for in the model results.

The input DEM is an artificially created U-shaped valley of 150 by 150 meters, with a cell size of 1.5 meters (Figure 1). The slopes facing north- and southward have a sinusoid form, and valley depth increases eastward, from 0 to 9 meters. Random noise of max 1 cm was added. The maximum slope is 12° (21%), which reaches the limit for agricultural use (Bibby and Mackney, 1969). The small cell size of 1.5 meters is required to simulate the effect of pit and mound topography created by tree throw on spatial infiltration patterns. The landscape was designed to display typical topographic features present in loess areas, but we exaggerated the spatial variation of slope positions to limit catchment size and reduce calculation time.

As parent material we chose a homogeneous loess without carbonates and a soil texture of 15% sand, 75% silt and 10% clay, which falls in the typical range of loess deposits (Muhs, 2007; Pécsi, 1990). We assumed an infinite loess thickness to avoid any effects of layers underneath with different lithologies. However, for computational reasons, we worked with an initial loess layer of 3 m with free leaching of water and dispersed clay at the lower boundary. This approach reduced the amount of soil layers and prevented numerical instability from the pedotransfer function for depth-dependent bulk density. The selected thickness left sufficient soil material so that the bottom of the loess was not reached by erosion during any of the model runs. The model requires a latitude to calculate solar inclination on the slopes. We selected the latitude of 50 degrees north, which is in the center of the range for loess occurrence reported by Muhs (2007, 40-60°N). We selected the rainfall scenarios based on most common rainfall in loess areas. For this, we made an overlay of a coarse resolution global loess map (Dürr et al., 2005) with a global annual rainfall map (Fick and Hijmans, 2017). The distribution of rainfall from the overlay showed peaks at ~600 and ~900 mm (Figure 1). We selected these annual quantities of rainfall as input for our scenarios and we added a scenario of 300 mm to capture a wider range of climates. The model requires as input daily data on rainfall [m], rainfall duration [h], and minimum, mean and maximum temperature [°C]. Rainfall amount is required to calculate how much water flows through the soil landscape. Rainfall intensity is required to determine whether and how much overland flow occurs, by comparing rainfall intensity with soil hydraulic conductivity. Rainfall intensity is calculated by dividing the rainfall amount by the daily duration [m hr$^{-1}$]. Temperature data is required to calculate potential evapotranspiration (Hargreaves and Samani, 1985). As we want to simulate general trends in soil and landscape evolution, we do not need site-specific data for the different scenarios. Instead, an arbitrary weather dataset was scaled to the total amount of rainfall from the different climate scenarios. We used weather data from the German weather station Grünow, which is located at 53.3°N, 13.9°E (DWD Climate Data Center (CDC), 2018a, b). The potential evapotranspiration is around 600 mm a$^{-1}$ for this dataset and is applied to all simulations. Combined with the rainfall scenarios, the scenarios can roughly be classified as dry (300 mm rainfall), humid (600 mm rainfall) and wet (900 mm rainfall). In the rest of this paper, we will use the terms dry, humid and wet to refer to the different rainfall scenarios.

We simulated the development of soils and landscapes for 15000 years, resembling the age of most post-glacial soils. In the first 14500 years of the simulations, soil and landscape development occurred under natural conditions and land cover. In the last 500 years of the simulations, we introduced agricultural land use by changing vegetation type and introducing tillage erosion. This duration was selected because it loosely reflects the onset of Medieval intense agriculture in many areas (Van der Meij et al., 2019) and should be seen as upper limit of onset of intensive tillage. Each of our simulations assumes a constant climate throughout the 15000 simulated years. Although we expect our model to be suitable to investigate the effects of a changing climate on soil and landscape evolution, this is beyond the scope of this study.

## 2.3    Analysis and evaluation

The model potentially outputs all soil properties for each layer at each location at each time step. Additionally, elevation change resulting from all processes at each location at each time step can be saved. In order to be able to interpret the results, we had to aggregate the results in several ways. We focused on select soil and terrain properties. The selected soil properties are soil organic matter stock [kg m$^{-2}$], which is the total amount of SOM in a soil column, and the depth to the Bt horizon [m], which we defined as the depth where the clay content first exceeds the initial clay fraction of the soil. The selected terrain properties are slope [degrees], topographic position index (TPI [m]), calculated with square windows 15*15 cells (22.5*22.5 m), and the topographic wetness index (TWI [-]). In most figures, we present two moments in time. These are the end of the natural phase (t = 14500) and the end of the agricultural phase (t = 15000). We present the results in the following ways:

- To show the development of soils and catenae, we show transects across the catchment (Figure 2), and plots of soil profile evolution, for three landscape positions and three rainfall scenarios (Figure 3);
- To compare natural and agricultural soil properties, we show catchment-averaged depth distributions of clay and SOM fractions (Figure 4).
- To show the impact of geomorphic processes on the terrain, we show cumulative elevation changes at the end of the natural and agricultural phase, and we show contributions to elevation change for each geomorphic process over time (Figure 5).
- To quantify the spatial heterogeneity of the selected soil and terrain properties, we calculated experimental semivariograms (Figure 6), using the gstat package in R (Pebesma, 2004). Experimental semivariograms give a measure of the variation between properties of soils as a function of distance between soils. We compared the semivariograms of depth to the Bt horizon with semivariograms made from field observations in a natural and agricultural site. The experimental semivariograms from the model results were calculated with a lag of 2 m, while the experimental semivariograms from the field data were calculated with a lag of 20 m.
- To visualize soil-landscape relations, we show how the selected soil properties and terrain properties are correlated and how these correlations change through time (Figure 7).

• To disentangle the effect of various factors on soil properties, we performed an analysis of variance (Table 3). We selected the depth to Bt and the carbon stock at the end of the natural and agricultural phase as dependent variables. As independent variables we selected climate [three rainfall classes], land cover or use [natural or agricultural], and landforms [three elevation classes with equal elevation ranges, representing plateau, slope and valley (Figure 1)].

## 3    Results

Here we present the results from the HydroLorica model. Section 3.1 shows the patterns, distributions and changes of soil and terrain properties in space and time. Section 3.2 shows the results from the statistical analyses to quantify and summarize spatial and temporal soil and terrain patterns. In the Supplements 2 and 3 we provided two animations to help visualize the simulated soil and landscape evolution. The animations show 1) maps of soil and terrain properties and forest cover and their changes through time, and 2) maps of elevation change by each geomorphic process and their changes through time.

### 3.1    Simulated soil and landscape evolution

The results of HydroLorica show clear differences in the development of soil profiles at different landscape positions, for the different rainfall and land cover/land use scenarios (Figure 2, Figure 3). In the natural phase, the forest cover shows a clear climatic and topographic dependence (animation in Supplement 2). For greater rainfall, there is a higher forest cover. The spatial pattern is mainly controlled by slope orientation. The north-facing slopes display a higher forest cover due to lower evapotranspiration. The valley and the hillslope depressions show a higher forest cover due to the higher moisture availability as consequence of surface runoff. Higher rainfall also leads to deeper eluviation of clay at each landscape position, showing more pronounced Bt horizons. Also, the soil profiles get more disturbed by tree throw with higher rainfall, as can be seen by the fluctuations in elevation and SOM stocks. The depth to the Bt horizon remains at the same position below the surface at the eroding position. At all locations, SOM stocks reach an equilibrium after ~3000 years, but most of the SOM is generated in the first 500 years.

In the agricultural phase, relief changes much faster, leading to truncation of the eroding soil profile (Figure 3). Also, SOM stocks decrease substantially in the soil profiles due to lower input. At the deposition site, there is a small increase in SOM stocks at the end of the agricultural phase, caused by the continuous input of soil material. The increased elevation change is well visible in Figure 2. After the natural phase, there is limited elevation change on the slopes, with some water erosion at the valley bottom forming a v-shaped gulley. After the agricultural phase, the hillslopes are heavily eroded, while the valley bottom is filled with colluvium. The high erodibility of clay that we simulated in the model affected the clay distributions in the model results. In the natural phase, topsoil clay gets laterally relocated from the hillslopes to tree throw pits and the valley bottom. This clay was partly replenished from the subsurface by bioturbation. This led to a net loss of clay from the entire depositional profile in the wet scenario, due to higher water flow and erosion potential (Figure 3). In the agricultural phase, clay does not get trapped in tree throw pits anymore, but leaves the catchment with the water. This reduced the clay contents even more at the valley bottom (Figure 2).

Figure 4 shows how clay and SOM fractions vary with depth throughout the entire catchment. The presented Probability Density Functions (PDFs) show multi-modal distributions of the soil properties, which cannot simply be captured using summary statistics. Both higher rainfall and agricultural land use increase the heterogeneity of clay profiles in the landscape, as can be seen by the wider ranges of the different PDFs throughout the entire depth profile. Also the occurrence of Bt horizons decreases with higher rainfall, due to losses of clay by lateral erosion rather than vertical transport as mentioned in the previous

paragraph. With higher rainfall, the percentages of soils with a Bt horizon occurring in the natural settings are 98%, 93% and 62%. For the SOM profiles, higher rainfall also leads to more heterogeneity. Especially in the topsoil a larger spread is simulated. Cultivation reduces the fraction and the topsoil variation, due to lower input and vertical and lateral topsoil homogenization (Figure 4 & Table 2).

All scenarios show a net elevation loss in the natural phase (Figure 5a). Creep transported hillslope material to the valley bottom, which water erosion partly removed from the catchment. The terrain becomes rougher with higher rainfall, due to increased water erosion and a higher occurrence of tree throw. Indirectly, the rougher terrain leads to increased creep rates, because of the locally increased relief gradients. Tillage erosion has had by far the largest impact on the terrain (Figure 5), overprinting the effects of natural geomorphic processes.

## 3.2    Statistical analysis of soil and terrain properties

Semivariograms summarize the spatial autocorrelation of soil and terrain properties as a function of distance between soil locations (Figure 6). The semivariogram contains three parameters. The nugget is the intercept with the y-axis, representing the local variability of the data and (in empirical studies) measurement uncertainty. The sill is the asymptote of the semivariogram and represents the maximum variability between pairs of observations at a distance where their proximity no longer matters. The range is that distance where the semivariogram levels off, approaching the sill. The range thus represents the maximum distance over which properties from two locations are autocorrelated.

In the natural phase, higher rainfall substantially increases the sill of soil and terrain properties regardless of distance; soils and terrain are thus more variable in space for higher rainfall, but do not display stronger spatial autocorrelation. Especially the SOM stock shows high semivariance over all distances in the wet scenario, due to a larger spatial redistribution by water. In the agricultural phase, the differences between the rainfall scenarios are much less pronounced; the variations in the properties are similar for each rainfall scenario. The local variation, expressed by the nugget, decreases in the agricultural phase because of short-range homogenization by ploughing. For the soil properties (Figure 6A&B), the range and sill general increase compared to the natural situation, while the topographic properties show sills and ranges similar to or lower than the natural settings. The differences in semivariance of the depth to Bt horizons in natural and agricultural settings appear also in semivariograms calculated from field data (Figure 6C). The data from Meerdaal (a natural forest in the loess belt in Belgium) shows a semivariogram that fluctuates around a constant value, while the data from agricultural field CarboZALF-D (located on glacial till in NE Germany) shows increasing semivariance with distance. The shapes of the field semivariograms match those of the model results, but note that the distances of the field data are five times larger than those of the model results, while the sills are about half.

The correlations between soil and terrain properties also differ between rainfall and land use options (Figure 7). In the natural phase, soil-landscape correlations are generally limited to 0.25, with exception of the correlation between depth to Bt and slope in the humid scenario. In the agricultural phase, the correlations initially increase for each combination of soil and terrain property, up to 0.8. The correlations generally approach constant values in the agricultural phase. An exception to these patterns

are the same correlations between slope and depth to the Bt horizon in the humid scenario. Those correlations increase to 0.4, and decline again in the agricultural phase. These large correlations in the natural phase appear from relatively little disturbance by tree throw and sufficient water to redistribute in the landscape. The small wiggles in the correlation lines are caused by minor uncertainties in our algorithm to derive soil properties from the model results.

Table 3 shows the results from the analysis of variance, which shows how much of the variance in soil properties at the end of the natural and agricultural phases can be explained by different factors (Table 3). The variance in depth to the Bt horizon can be partly explained by rainfall (18%) and landscape position (23%), when considering all data together. However, the largest part of the variance remains unexplained. For the SOM stocks, most of the variance can be explained by the land use (72%). When grouped per land cover/use, about half of the variance of depth to Bt can be explained by either rainfall (natural phase)

or landform (agricultural phase). For the SOM stocks the dominant factors are the same, but the variance in the natural soil-landscape can only be partly explained by rainfall (14%) and a large part remains unexplained.

# 4 Discussion

## 4.1 Soil patterns and properties

### 4.1.1 Soil patterns

Soils have been affected by humans for over thousands of years, either directly by agricultural use, or indirectly by adjusting factors that form the soil, such as vegetation or climate (Amundson et al., 2015; Bajard et al., 2017; Dotterweich, 2008; Stephens et al., 2019). Therefore it is difficult, if not impossible, to find locations where truly natural soils can be observed and compared to agricultural soils in similar settings. Model simulations enable this comparison, as we show in this study. Unfortunately, there is limited field data to calibrate and validate the model. To our knowledge, the dataset from Vanwalleghem et al. (2010) is the only dataset that enables quantification of the spatial distribution of natural soils and link it to terrain properties at a local to regional scale, similar to the setting we simulated. In this Section, we rely mainly on this dataset to discuss and evaluate the patterns of natural soils we simulated with our model. For the agricultural soil patterns, we use an extensive dataset from an intensively managed agricultural field in northeastern Germany (CarboZALF-D, Van der Meij et al., 2017). In our model simulations, we simplified the agricultural conversion by assuming a single vegetation type in the entire catchment and direct intensive management with tillage. This enabled us to isolate the role of tillage erosion on the development of agricultural soil and landscape patterns. We did not consider a slow historical development of the agricultural system with increasing management intensity and upscaling of agricultural field sizes. The results of our simulations should be considered as within-field variation in soil and landscape properties. In smaller scale farming, the within-field soil-landscape relations will also be present, but they are probably secondary to variation between fields caused by different management (history), vegetation type or anthropogenic structures such as hedges, banks and roads (e.g. Follain et al., 2006; Peukert et al., 2016; Yemefack et al., 2005).

We used semivariograms to illustrate the spatial autocorrelation of soil and landscape properties (Figure 6). Semivariograms are very case study-specific, because the range, sill and nugget are affected by the scale of topographic and lithogenic variation, different rates of pedogenic and geomorphic processes and different types of human disturbances in the landscape. Therefore, we only compare the trends in the semivariograms from model and field results to evaluate the type of spatial autocorrelation of soil properties in such settings.

Figure 6B&C shows experimental semivariograms of depths to Bt horizons in model and field data. In both panes, the agricultural settings show higher spatial autocorrelation compared to the natural settings, expressed by the higher sill and range. This indicates that in agricultural fields the depths to Bt horizons are more spatially organized (higher large scale variability), with larger differences between different landscape positions. In natural areas, the spatial differences in depth to Bt horizon are lower and there is less spatial organization of the depth distributions. The model and field results show different magnitudes in nugget, range and sill. This can be explained by 1) the high density of data points in the model results which enabled us to calculate the semivariance over very short distances reducing the nugget, and 2) the fact that we used a very condensed DEM with high local variation in topographic properties as input for the model results which led to high local

variation in soil properties too. Nonetheless, the similar trends in the field and model semivariograms indicate that the general soil patterns from model and field results agree. Also, the correlations between soil and landscape properties are similar for field and model results. Vanwalleghem et al. (2010) found correlations between different horizon depths and topographic properties with $R^2$s ranging between 0.02 and 0.1, which are the same order as most correlations we calculated in Figure 7.

These similarities indicate that our model HydroLorica simulated the essential processes that form these natural soil patterns. Our simulations show a large diversity of natural soil patterns, influenced by the amount of rainfall and associated vegetation type. The available water leads to a regionally higher rate of soil development, for example in the form of deeper clay eluviation (Figure 3), and also to a greater lateral redistribution of soil material by water erosion and tree throw (Figure 5) and spatially varying infiltration rates. With more rainfall, the higher rates and interactions between these processes lead to a spatially more

heterogeneous soil pattern, as expressed in higher ranges and sills in the semivariograms (Figure 6). This local variation in pedogenesis due to different water input has been recognized and partly accounted for in other modelling studies (Finke et al., 2013; Saco et al., 2006; Shepard et al., 2017), but had not emerged from soil-landscape evolution studies. Also the terrain, summarized by slope, TPI and TWI, becomes more heterogeneous with higher rainfall. Water flow thus affects soil and terrain patterns in a similar way.

Intensively managed agricultural soils display entirely different patterns compared to natural soils. There is lower small scale variability due to the absence of tree throw and local homogenization by tillage, while the semivariograms of soil properties suggest higher sills, i.e. higher large scale variability and spatial autocorrelation of soil properties compared to natural soil properties. This is due to the slope-dependent intensity of tillage erosion (Phillips et al., 1999). This erosion leads to truncation of soils at convex positions, while concave positions have a net accumulation of material (De Alba et al., 2004). This truncation

is visible in many agricultural landscapes, because subsurface horizons with different colors get exposed at the surface on heavily eroded locations (e.g. Smetanová, 2009; Van der Meij et al., 2017). In contrast, terrain properties seem to display lower spatial variation in agricultural landscapes. The smoothing effect of tillage on the terrain removed local pits and rills created in the natural phase. We hypothesized earlier that a smoother terrain would have higher hillslope connectivity, leading to increased water erosion (Van der Meij et al., 2017). However, we observed the contrary in our model results (Figure 5). The

export of sediments from the catchment might be higher, but the uptake and local redistribution of sediments on the hillsope is lower, because local steep gradients are removed. Tillage is thus the dominant process forming agricultural soil patterns. The effect of anthropogenic soil erosion on soil heterogeneity far exceeds effects of changes in for example rainfall, which shows the huge impact we have as humans on soil-landscape development.

### 4.1.2   Process calibration and verification

The rates of the simulated processes were difficult to calibrate and validate. This is mainly due to a lack of field data that covers a range of climatic, topographic, chronologic and geographic settings (Van der Meij et al., 2018). Such data are essential for formulating pedogenic functions that are applicable in a wide range of settings instead of only in case studies, or for verifying model results. The chronosequence collection of Shepard et al. (2017) is a global dataset of soils in various settings

covering different time steps. This dataset could be a good starting point for developing such functions owing to its large coverage. But as chronosequences are generally situated in relatively flat, stable landscapes, they often do not contain information about variations of soil properties at small distances, as function of local terrain (Harden, 1988; Sauer, 2015) – with the exception of some pro-glacial soil chronosequences whose use is limited because of their extreme climate and parent material (Egli et al., 2006; Temme and Lange, 2014). Such more complete information is essential for understanding the formation of soil patterns, as illustrated in the previous Section. Therefore, we suggest to include topographic variation in future chronosequence studies (Temme, 2019). A dataset covering different geographies could also raise the comparison of model and field results beyond the case study-level.

In this study, we worked with an artificial landscape to avoid effects of uncertainties and local variations in initial and boundary conditions that are often present in data from field settings (e.g. Van der Meij et al., 2017). This allowed us to investigate the universal effects of changes in rainfall and land use in the model results, as a function of terrain morphology. Although uncertainties in boundary conditions appear to have a limited effect on the outcomes of soil evolution models, uncertainties in initial conditions can strongly influence the results (Keyvanshokouhi et al., 2016).

One soil property for which there is plenty of data on the spatiotemporal variation is soil organic matter or carbon, due to the current interest in its potential to store atmospheric carbon (Minasny et al., 2017). We used a regional dataset from the loess plateau to calibrate our SOM cycle in agricultural landscapes and we used carbon sequestration rates for adjusting the SOM balances for forest and grassland areas. The modelled SOM stocks for agricultural sites match the field data fairly well (Table 2), but stocks for natural areas are estimated higher than often observed. For example, in Bavaria, Germany, carbon stocks in the first meter, including the optional litter layer, are 9.8-11.8 kg m$^{-2}$ (Wiesmeier et al., 2012), where we simulated 15.7-17.1 kg m$^{-2}$ in our natural settings without consideration of a litter layer. Also the depth distributions are different. De Vos et al. (2015) found that 50% of the carbon stock occurs in the top 20 cm in European forests on various parent materials. In our results this is around 20%. This implies that agriculturally-derived SOM depth functions are not suitable to calibrate natural SOM depth functions, probably because input, vertical redistribution, litter quality and decay of SOM behave differently in natural and agricultural sites. To calibrate these parameters, data from agricultural and natural sites in close vicinity are needed, to avoid effects of geographic and climatic differences. We are currently not able to simulate and calibrate these processes properly.

## 4.2    Drivers of soil formation

### 4.2.1    Soil forming factors

Different soil forming factors dominate the variance in soil properties in natural and agricultural systems (Table 3). In natural systems, rainfall is the dominant factor explaining the variance. In scenarios with greater rainfall, rates of soil and landscape change are larger, leading to more complex patterns. Although we did not simulate a changing climate, the results suggest that we can expect more stable conditions with similar pedogenesis rates throughout the landscape in periods with lower rainfall,

while periods with greater rainfall may induce landscape change and spatially varying rates of pedogenesis. The major driver for this increased landscape change is the higher occurrence of tree throw. The higher water availability increases forest cover, leading to more tree throws (see animations in the Supplements 2 and 3).

Although our vegetation module is very simple, it was able to simulate the climatic and topographic control on vegetation patterns which affect geomorphic and pedogenic processes. We would expect similar results to be obtained if a more complex vegetation module that does justice to ecological complexity (i.e. resilience, succession) would be incorporated.

In intensive agricultural systems with large fields, landform is the dominant factor explaining the variance (Table 3). This shift from external factors in natural systems to internal factors in agricultural systems marks the importance of geomophic processes on agricultural soil patterns. Although relief controls rates and directions of geomorphic processes, the type of process is human-controlled. Humans have a massive impact on soil development (Amundson and Jenny, 1991; Dudal, 2005). Direct effects include agricultural use, excavations, introduction of organisms and creation of new parent materials (Richter et al., 2015), while indirectly anthropogenic changes in climate can have severe effects on soil properties (Nearing et al., 2004; Schuur et al., 2015). We have focussed on the main of these anthropogenic changes in loess landscapes: removal of forest and complete introduction of tillage, even though intermediate forms with incomplete clearing, smaller fields and forested borders may have historically existed. Humans as soil forming factor form new catenae (anthroposequences) and soil patterns, where the ultimate pattern only depends little on the initial variation (Figure 6). In our model results, we observe four of the six anthropogenic changes to soils, as described by Dudal (2005): human-made soil horizons, deep soil disturbance, topsoil changes and changes in landforms. These changes substantially affect soil functions, such as biodiversity and food security. Our simulations thus support that humans are the dominant factor for forming soils in agricultural landscapes.

### 4.2.2 Soil-landscape (co-)evolution

The development of soils and landscapes is not merely a collection of individual processes, but also of interactions between different processes. When processes interact, and when changes to soils and landscapes are in the same order of magnitude, soil-landscape co-evolution can occur. This co-evolution can amplify or diminish certain processes, or can completely change the direction of soil and landscape evolution (Van der Meij et al., 2018). Often, co-evolution is used to describe soil and landscape processes with similar rates, but that do not necessarily interact (e.g. Willgoose, 2018). This would imply that these processes would co-occur rather than co-evolve. In this Section we evaluate some co-occurring processes in HydroLorica to see whether co-evolution occurred. There are different co-occurring processes in the natural phase of slow landscape change compared to the agricultural phase of intense landscape change.

#### 4.2.2.1 Lateral and vertical transport

We will first consider vertical and lateral soil transport processes. Soils and hillslopes can be considered as a series of transport ways or conveyor belts (Román-Sánchez et al., 2019). Vertical transport or mixing occurs by bioturbation including tree throw and clay translocation, whereas lateral transport occurs by creep, tree throw, water erosion and tillage erosion. Interactions

between processes can occur where transport ways affect the same material. Two examples we will discuss here are the vertical and lateral transport of clay and the interaction between creep and water erosion in the valley bottom.

The vertical translocation of clay is simulated in our model by an advection-diffusion equation, where the advective part is the downward transport by water flow and the diffusive part a homogenization by bioturbation (Jagercikova et al., 2017). When the rates of advection and diffusion are equal, the upward transport of clay by bioturbation equals the amount of downward translocation by water; the clay-depth profile of the soil occurs in steady state and will not change substantially. Steady-state circumstances are however rare in natural soil systems (Phillips, 2010). Our simulations do not show steady-state
circumstances, because in our simulations there is always lateral transport of soil material that continuously changes slope and terrain properties and affects the soil's clay balance, complicating the achievement of a steady state. Periodic water erosion can remove substantial amounts of clay that have been transported to the surface by bioturbation. This is well visible in the results of the wet scenario (P = 900 mm), where only 62% of the soils developed a Bt horizon. The other 38% had insufficient clay left to be classified as Bt according to our criteria. These results are quite extreme for such a small catchment as ours,
probably due to too high simulated rates of water erosion, but they do show how pedogenic and geomorphic processes can interact in sloping terrain. In the natural phase the rates of clay translocation are similar to those of geomorphic processes. The recovery of the clay-depth profiles after disturbance of e.g. tree throw takes similar times (~1000s of years) as the re-occurrence of a sequential tree throw event in the vicinity (Figure 3). Tree throw also temporarily changes rates of clay translocation by concentrating infiltration in the created pits. In the agricultural phase the rates of geomorphic processes far exceed the rates of
clay translocation. This causes truncation of the soils, exposing the Bt horizons at the surface, and burying these horizons elsewhere in the landscape. The clay profiles at eroded sites do not have time to react to the geomorphic disturbances. However, clay illuviation can start as a new pedogenic process in older depositional areas (supplementary information of Leopold and Völkel, 2007; Van der Meij et al., 2019; Zádorová and Pení žek, 2018).

Another interaction that emerged from the simulations occurred at the valley bottom. Soil creep transported hillslope material
downslope, whence the concentrated water flow in the valley removed it from the catchment, creating a v-shaped valley bottom (Figure 2). This constant removal of material maintained the gradients that were used by soil creep to deliver new material. This interaction can be observed in various small hillslope catchments, which display typical v-shaped gullies in the valley bottoms (e.g. Swanson and Swanston, 1977; West et al., 2013). Although this is not an interaction between pedogenic and geomorphic processes, it determines to a large extent how soil material gets redistributed along a hillslope and eventually gets
exported from the catchment. In the agricultural phase, diffusive transport in the form of tillage erosion dominates over advective transport by water. As a consequence, the typical v-shapes fill up and are replaced by u-shaped valleys. These valley fillings consist of coarse material from which most clay was eroded (Figure 2). In agricultural areas, such infillings can temporarily remove erosion gullies, but due to local water availability, they remain weak spots for future water erosion (Poesen, 2011).

 #### 4.2.2.2    Soil organic matter dynamics

Rates of SOM accumulation and decomposition far exceed rates of clay translocation. SOM stocks recover quickly after a disturbance by tree throw and can keep up with intense landscape change by tillage (Figure 3). Freshly exposed, reactive soil material at eroding sites quickly accumulates new SOM, whereas SOM gets buried at depositional positions. Meanwhile, SOM decomposition increases during transport (Doetterl et al., 2012). In our simulations, the SOM stocks decrease substantially in the agricultural phase, mainly due to lower SOM input (Figure 3). Carbon stocks show relatively homogeneous distributions throughout the catchment (Figure 4), despite large spatial differences in erosion and deposition. This indicates that landscape change in both natural and agricultural systems did not induce substantial heterogeneity in SOM stocks. The small differences in SOM stocks in agricultural settings depend on landform (Table 3). These differences mainly emerge from differences in soil thickness at erosion and deposition positions. Deposition positions show a slight increase in SOM stocks after cultivation, while erosion positions show continually decreasing SOM stocks (Figure 3). The differences in SOM stocks in the model results are thus related to burial of colluvium in the valley bottom. SOM cycling is heavily influenced by erosion processes, but erosion rates do not depend on the SOM cycling. In tillage-dominated systems, erosion rates do not depend on SOM content or SOM dynamics in the soil. The co-occurrence of SOM cycling and tillage erosion in agricultural settings thus does not lead to co-evolution.

The interactions between erosion and the SOM cycle are currently under debate, especially whether agricultural redistribution provides a carbon source or sink through affecting biogeochemical cycles and exporting carbon from fields and catchments (Berhe et al., 2018; Chappell et al., 2015; Doetterl et al., 2016; Harden et al., 1999; Lal, 2019; Lugato et al., 2018; Van Oost et al., 2007; Wang et al., 2017), which shows the importance of considering landscape processes in pedogenic studies and vice versa. Moreover, intensive agriculture has been practiced for over 1000s of years in parts of the world (Stephens et al., 2019), emphasizing the need to consider centennial to millennial time periods in studies on anthropogenic forcing on soil systems.

#### 4.2.2.3    Did co-evolution occur?

The co-occurrence of processes does not necessarily implicate co-evolution. The analysis in this Section showed that soil and landscape processes co-occurred in both natural and agricultural settings, but that interactions between processes only occurred in natural settings. Rates of soil and landscape change are controlled by drivers such as water availability and vegetation type, and these drivers are influenced by soil, landscape and climate properties. Changes in one domain in the landscape have effects on the formation of all other domains. These interactions, or co-evolution, occur on both short and long timescales in the natural system. There are already considerable differences between the soil patterns from each scenario after 500 years of natural soil formation, due to the role of water and vegetation in soil-landscape co-evolution. These differences become more pronounced over time, due to progressive soil and landscape formation (Supplement 2).

In comparison, the differences between the patterns of each scenario after 500 years of agricultural land use are much smaller (Supplement 2). This is because anthropogenic processes such as tillage erosion occur at such high rates that most natural processes cannot keep up, and lead to more similar soil landscapes. In settings with uniform parent material such as we simulated, anthropogenic processes do not show co-evolution, because the rates of for example tillage erosion far exceed any

rates of natural soil and landscape change (Figure 5) and the rates of the anthropogenic processes are not influenced by soil properties. Tillage can introduce new processes or accelerate other processes e.g. by breaking up aggregates. However, these processes do not affect the rate at which a plough transports sediments through a landscape. If interactions between processes do not occur on shorter timescales, they will also not emerge over longer timescales as is the case with natural processes as described before. The occurrence of possible co-evolution of soils and landscapes thus depends on the type of processes that affect the system, not on the duration over which these processes change soils and landscapes. In other words, co-evolution is not time-dependent, but process-dependent.

Co-evolution of soils and landscapes can also occur via intrinsic thresholds which do not depend on changes in external drivers such as rainfall and land use. An example is the development of stagnating layers in the soil, which change the subsurface partitioning of water and can introduce reducing conditions. But, as we explain in Van der Meij et al. (2018), such intrinsic thresholds can currently not be modelled, because we lack the methods for estimating accurate soil hydraulic properties which drive this threshold behavior. Ideally, a model shows such threshold behavior without explicitly incorporating these thresholds in the model code as such imposed hard thresholds can cause problems when calibrating the model by creating sharp discontinuities in the model results as a response to slight variations in parameters (Barnhart et al., 2019). For these reasons we focused on heterogeneity and (co-)evolution related to external drivers in this research.

The soil and landscape interactions in natural settings emphasize the need of studying natural soil formation in a landscape context rather than a pedon context. Only when landscape are stable, flat and free of trees, changes in soil properties are not influenced by changes in terrain. In such settings, a 1D soil profile evolution model would suffice to simulate soil development in different landscape positions (Finke, 2012; Minasny et al., 2015). When rates of geomorphic processes far exceed those of pedogenic processes, for example in tillage-dominated systems, a landscape evolution model would suffice (e.g. Temme et al., 2017). In undulating landscapes where various hillslope processes occur, soils should be considered 3D bodies and soil-landscape evolution models are essential to simulate spatial drivers of soil and landscape evolution (Willgoose, 2018).

### 4.3    Predictability of soil patterns

In digital soil mapping, empirical relations between soil properties and their environment are used to predict soil properties through space (McBratney et al., 2003). In order to predict soil properties with environmental variables, the environmental variables should show variation over the same spatial scale as the variable to be predicted. On a hillslope scale, this variation often occurs in terrain properties (Gessler et al., 2000), while external factors such as climate often do not vary spatially at these scales. The shift from dominant external to dominant internal soil forming factors in explaining variance in observed soil properties (Table 3) thus has large implications for our ability to predict and map soil patterns. Human activity has created soil-landscapes that are well-suited for digital soil mapping. The correlations between simulated soil and several terrain properties all give the same signal (Figure 7): the correlations in the natural phase are limited, but increase rapidly in the agricultural phase. The switch from a natural to agricultural phase thus increases soil heterogeneity, but also soil predictability, which can be used to predict the soil properties in large-field settings. One should be careful extrapolating soil-terrain

relationships from agricultural areas to natural areas, as these correlations depend on land management and can give wrong results under different land cover.

Digital soil mapping performs well when predicting the spatial distribution of agricultural soils, but their applicability in time

is limited because of limited temporal data (Gasch et al., 2015; Grunwald, 2009). The limited observations in space and time can be supplemented or extrapolated by incorporating biogeochemical process descriptions to improve DSM (Angelini et al., 2016; Christakos, 2000, pp. 22; Heuvelink and Webster, 2001). However, the response of soils and terrains to changes in soil forming factors takes longer (decades to millennia) than the timespan over which we have observations (days to decades). Process-based models thus become increasingly essential for understanding how soils might change under projected scenarios

of land use and climate change (Keyvanshokouhi et al., 2016; Opolot et al., 2015), and HydroLorica shows a promising first example of such a model on a landscape scale that responds to changes in all five soil-forming factors, and by extension the human control on these factors.

# 5    Conclusions

Soils undergo substantial changes in the transition from a natural land cover to agricultural land use. Although these changes can be described conceptually, quantitative data to describe the changes in soil pattern are scarce. We developed a soil-landscape evolution model, named HydroLorica, which is able to simulate the evolution of soils and landscapes in both natural and agricultural settings, by simulating spatially varying infiltration as driver of soil formation and by inclusion of essential natural and agricultural processes such as soil creep, tree throw and tillage. We used this model to simulate soil and landscape

development in varying climatic settings, under changing land use, to quantify changes in variation and predictability of soil patterns. We reached the following conclusions:

- Natural and agricultural landscapes display different soil patterns. Natural soil patterns are more chaotic and random with higher precipitation. Their formation is dominated by local processes such as tree throw and spatially varying infiltration. Soil patterns in intensively managed fields are dominantly formed by tillage erosion processes.

Also, agricultural soil properties show larger correlations with terrain properties.

- In natural systems, rainfall is the main factor influencing soil variation. In agricultural systems, landform explains the largest part of variation. The most important factor affecting total soil variation is the human factor. Agricultural land use increases erosion rates, which changes soil patterns and creates and amplifies the topographic dependence of soil properties.

- In natural and agricultural settings there are different sets of processes that change soils and landscape with similar rates. In natural systems, these processes often interact and amplify or diminish each other, leading to soil-landscape co-evolution. In agricultural systems, these interactions are often missing and processes co-occur rather than co-evolve.

- Agricultural soil patterns in a large-field setting are easier to predict than natural soil patterns, due to the shift from

dominant external to internal factors that explain soil variation, which manifests itself in larger correlations between soil and terrain properties.

Soil-landscape evolution models are increasingly equipped to simulate soil landscape development in a variety of settings. Our contribution shows the added value of using water availability as spatially varying driver of pedogenesis to simulate soil and landscape development in natural settings. These developments are essential to study the vulnerability and resilience of soil

systems under the increasing pressure from land use intensification and the changing climate, but can also assist in understanding the long-term effects of management strategies such as reduced tillage or no-till on soil properties such as carbon stocks.

## 6    Model availability

Model code is available on request via the corresponding author.

## 7    Supplementary information

We provided the following supplements:

- Supplement 1: Model equations and parameters (document)
- Supplement 2: Maps of soil and terrain properties through time (animation)
- Supplement 3: Maps of elevation change due to the different geomorphic processes through time (animation)

## 8    Author contribution

The authors contributed to experimental concept and design, model development, data analysis and paper preparation in the following proportions: WMvdM (25%, 75%, 75%, 65%), AJAMT (25%, 25%, 15%, 20%), JW (25%, 0%, 10%, 10%), MS (25%, 0%, 0%, 5%). The authors declare that they have no conflict of interest.

## 9    Acknowledgements

We thank Tom Vanwalleghem (University of Cordoba, Spain) for sharing the Meerdaal dataset.

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

**11  Tables**

**Table 1: Overview of processes simulated in HydroLorica, including driving soil forming factors in the model, and landscape variable that is affected by each process. Humans are considered as additional soil forming factor (Amundson and Jenny, 1991; Richter et al., 2015).**

| Process | Abbreviation | Soil forming factor | | | | | Landscape variable affected | | | |
|---|---|---|---|---|---|---|---|---|---|---|
| | | **Cl**imate [rainfall] | **O**rganisms [vegetation type] | **R**elief | **P**arent material [soil texture] | **H**umans | Topography | Soil properties | Water balance | Vegetation type |
| Bioturbation | BT | | X | | | | | X | | |
| Carbon accumulation and breakdown | CAB | | X | | | | | X | | |
| Clay translocation | CT | X | | | X | | | X | | |
| Creep | CR | | X | X | | | X | X | | |
| Pedon scale water partitioning | WP | X | | | X | | | | X | |
| Surface flow | SF | X | | X | | | | | X | |
| Tillage | TI | | | | | X | X | X | | |
| Tree throw | TT | | X | X | | | X | X | | |
| Vegetation selection | VS | X | | | | X | | | X | X |
| Water erosion | WE | X | X | X | X | | X | X | | |

 **Table 2: Model and field organic carbon stocks (kg m-2) for different depth ranges, averaged over the catchment (average ± standard deviation). The model results were converted from SOM to SOC by multiplying the SOM stocks with 0.58 (Wolff, 1864).**

| Scenario / Depth range | Natural phase (t 14500) | | | Agricultural phase (t 15000) | | | (Liu et al., 2011) |
|---|---|---|---|---|---|---|---|
| | Dry (grassland) | Humid (mixed) | Wet (forest) | Dry | Humid | Wet | |
| 0-0.2 | 4.7±0.3 | 4.6±0.9 | 4.1±2.1 | 2.9±0.1 | 2.9±0.3 | 2.8±0.4 | 3.0±1.9 |
| 0-0.4 | 8.7±0.4 | 8.5±1.1 | 7.8±2.8 | 5.5±0.2 | 5.4±0.5 | 5.3±0.6 | 5.4±3.2 |
| 0-1 | 17.1±0.4 | 16.8±1.2 | 15.7±3.5 | 10.9±0.6 | 10.8±0.8 | 10.6±0.8 | 8.8±4.4 |
| 0-2 | 24.1±0.4 | 23.7±1.3 | 22.3±3.8 | 15.7±1.2 | 15.6±1.2 | 15.4±1.1 | 14.5±5.2 |
| Complete profile | 27.7±1.3 | 27.1±1.7 | 25.3±8.5 | 18.6±16.1 | 18.4±9.1 | 17.8±10.1 | - |

**Table 3: Results from the analysis of variance, indicating the proportion of variance in soil properties explained by the different soil forming factors. The data is both considered in total, and grouped per land use (natural or agricultural). The bold numbers indicate the largest part of the variance, either explained by one of the factors, or unexplained. All responses are significant (p<0.05).**

| | Depth Bt | | | SOM stock | | |
|---|---|---|---|---|---|---|
| | Total | Natural | Agricultural | Total | Natural | Agricultural |
| Rainfall | 0.18 | **0.49** | 0.08 | 0.02 | 0.14 | 0.02 |
| Landform | 0.23 | 0.04 | **0.51** | 0.04 | 0.01 | **0.56** |
| Land use | 0.01 | - | - | **0.72** | - | - |
| *Unexplained* | ***0.58*** | *0.47* | *0.41* | *0.22* | ***0.85*** | *0.42* |

## 12    Figures

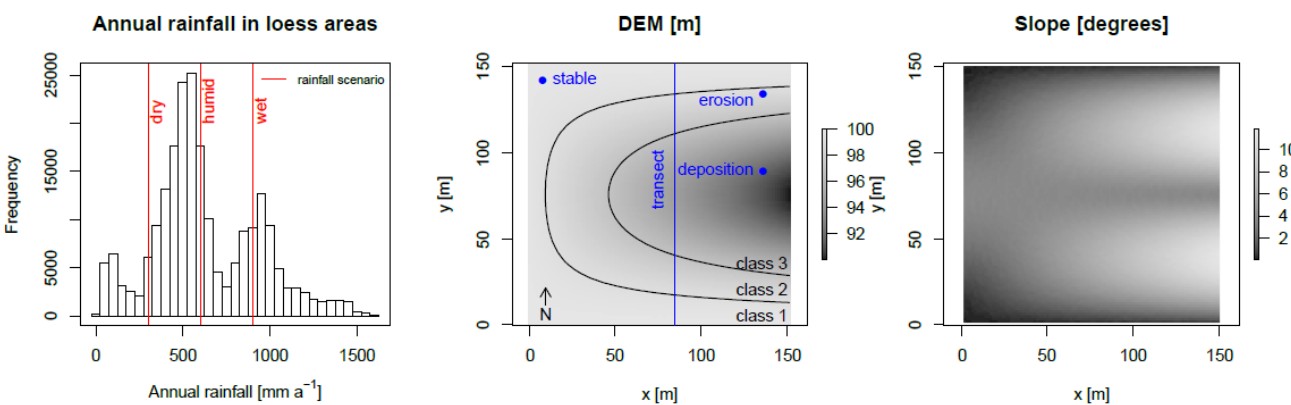

**Figure 1: Left: Annual rainfall in loess areas, derived from WorldClim. Red lines indicate the rainfall scenarios in this study: 300 (dry), 600 (humid) & 900 (wet) mm per year. Right: Maps of input DEM with corresponding slope map. Extent of the DEM is 150*150 meter, with a cell size of 1.5 meter. The different classes indicate elevation classes used in the ANOVA (Table 3). The blue dots and line indicate the location of the soil profiles and transect displayed in Figure 2 and Figure 3.**

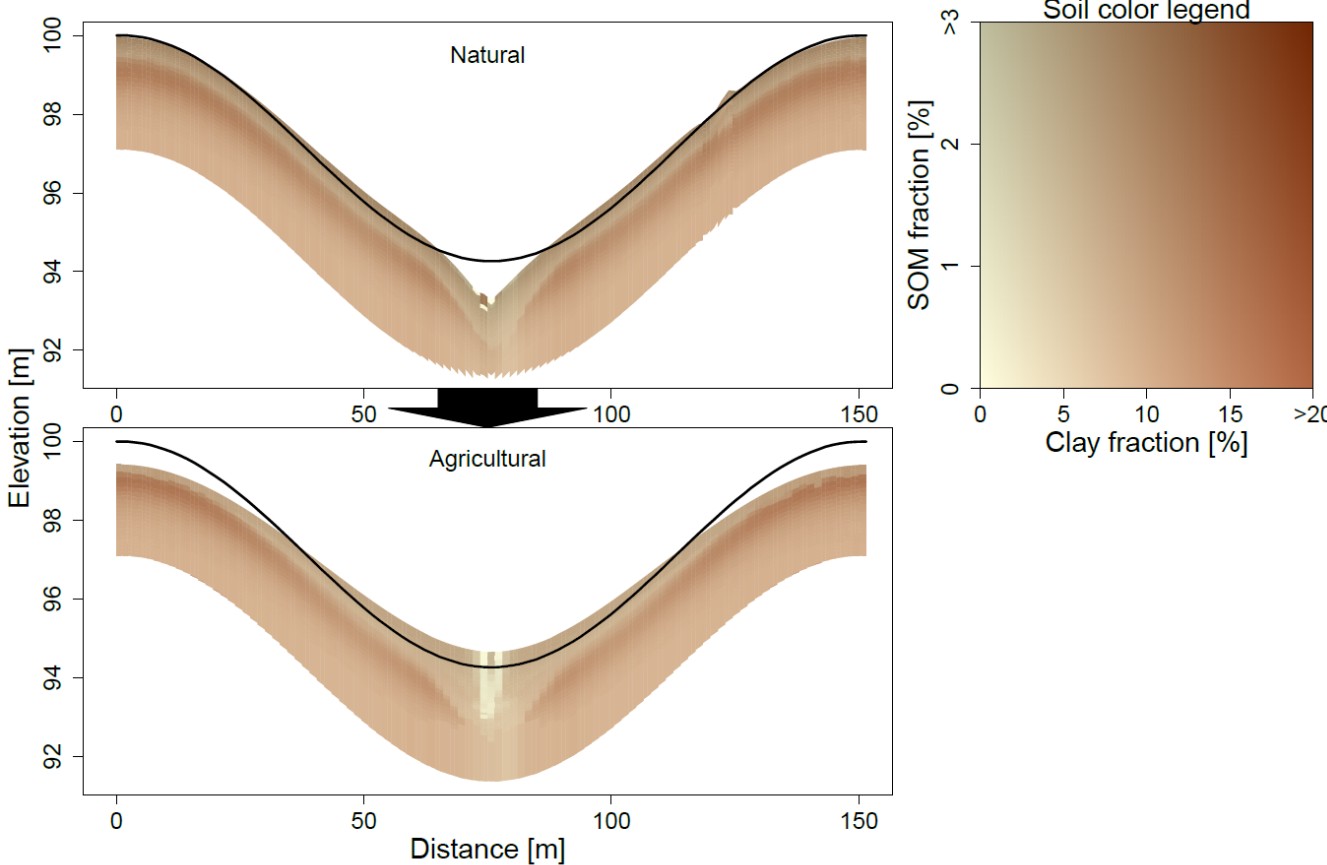

**Figure 2: Transect through the catchment at the end of the natural phase and the end of the agricultural phase for the humid scenario (P = 600 mm). The black line indicates initial topography. See Figure 1 for location of the transect.**

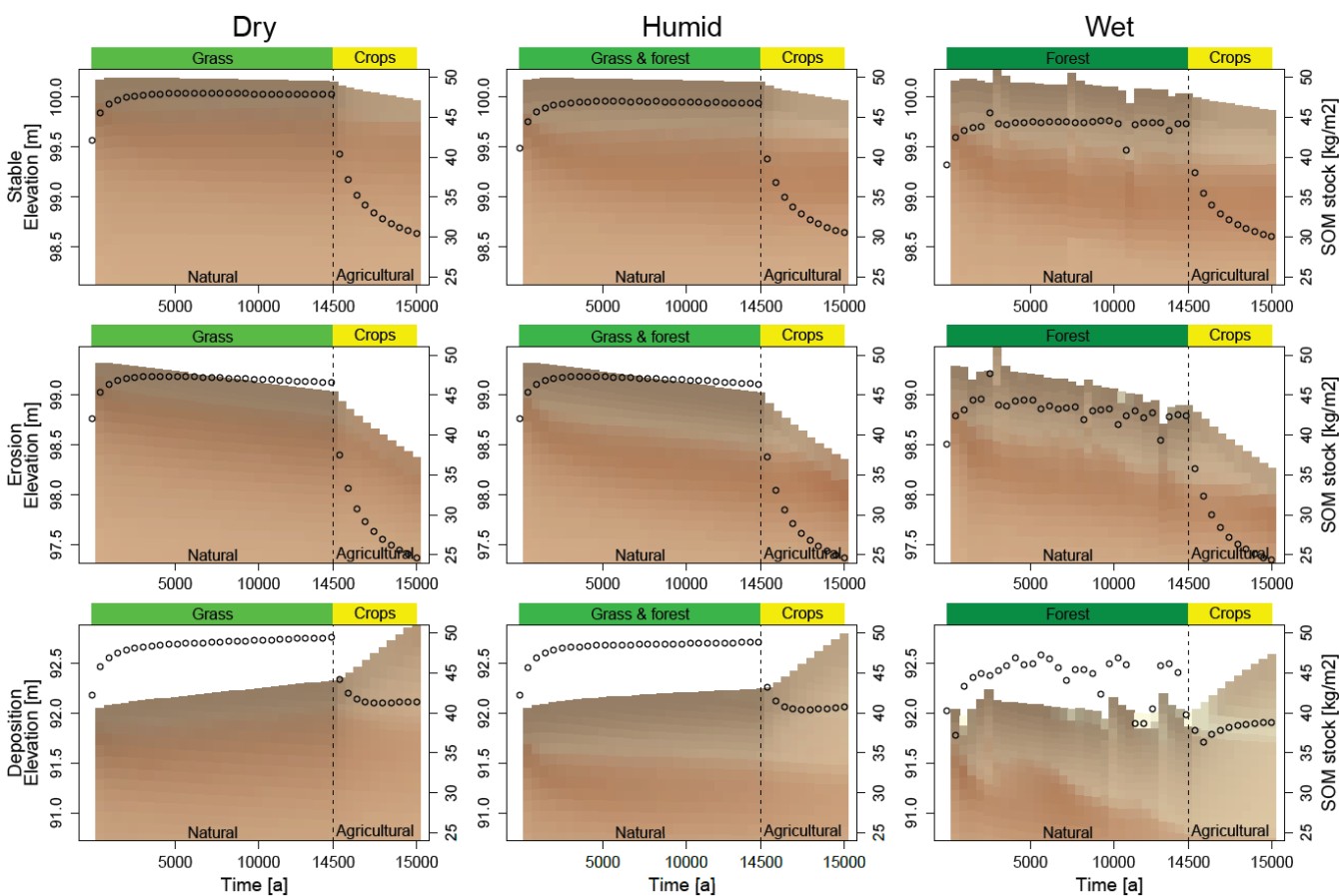

**Figure 3: Evolution of soil profiles through time (x-axis) on a stable, eroding and depositing position (rows), for the different rainfall scenarios (columns). The colored bars atop the plots indicate land cover (natural) and land use (agricultural). The points indicate the SOM stocks (right y-axis). Note that the natural and agricultural system have different x-axes scales to visualize both systems. In the agricultural system, an observation is shown each 500 years. In the agricultural phase each 50 years. See Figure 1 for locations of the soil profiles. See Figure 2 for the soil color legend.**

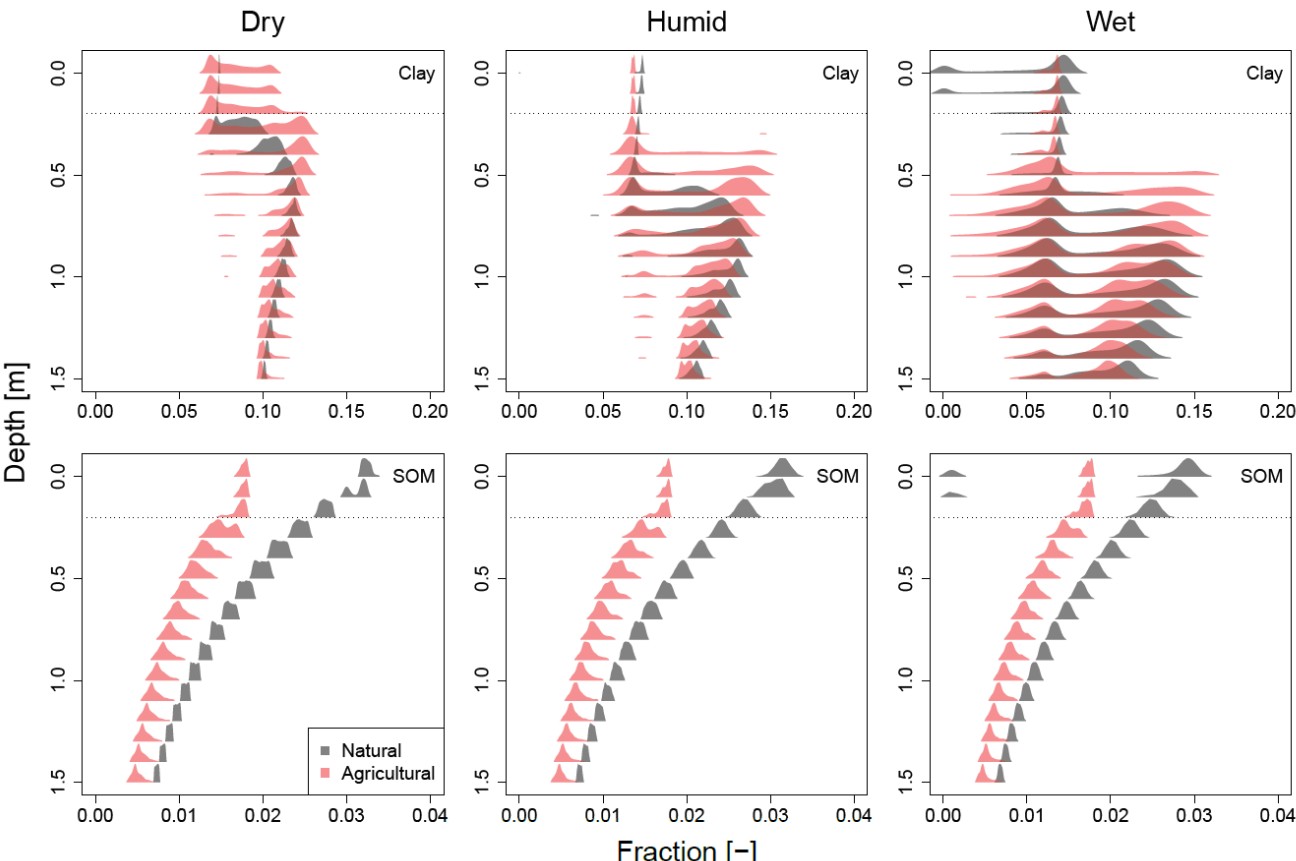

**Figure 4: Probability Density Functions (PDFs) showing the multi-modal distributions of soil properties throughout the catchment per 10 cm depth increment. We only show probabilities larger than 5% for clarity. The presented soil properties are clay fraction (top) and SOM fraction (bottom), for the different rainfall scenarios (columns). Grey colors represent the natural soils, while red colors represent agricultural soils. The horizontal dotted line indicates the ploughing depth used for simulations (20 cm).**

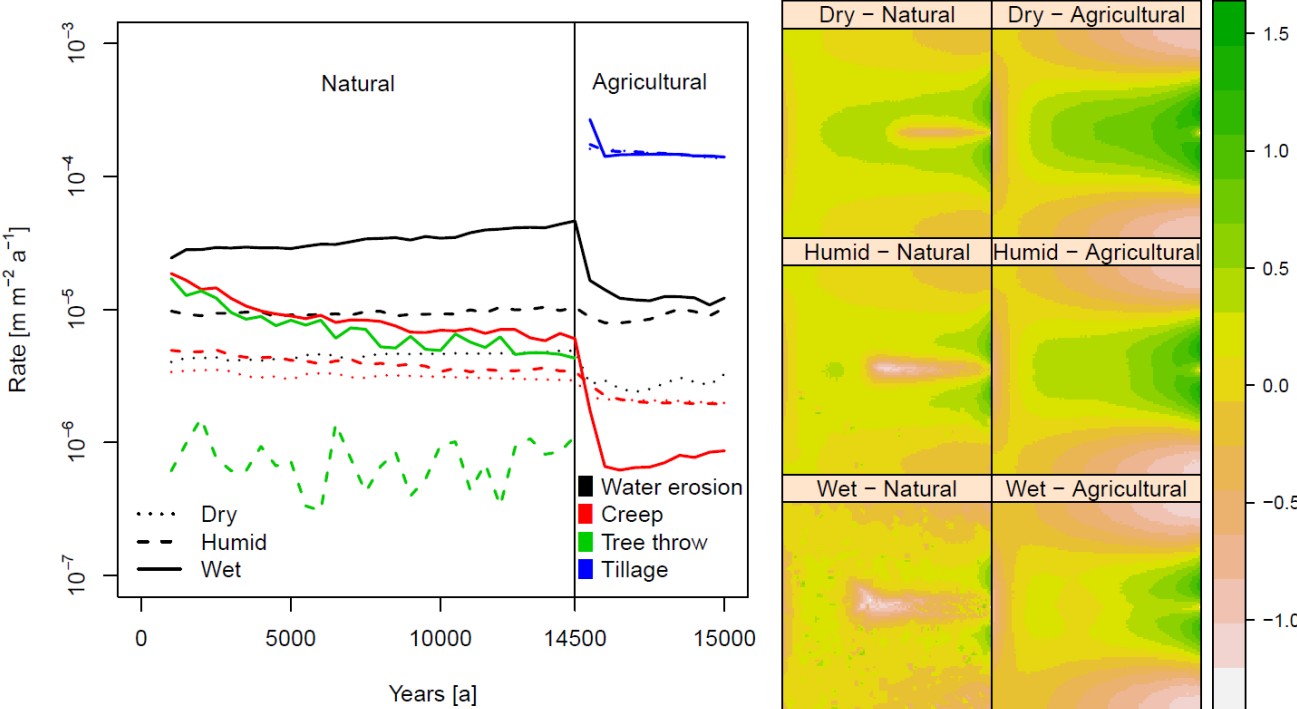

**Figure 5: Left: average erosion rates throughout the catchment for the different geomorphic processes over time. The colors represent different geomorphic processes, and the line types represent different rainfall scenarios. Note that the y-axis is log scaled. Right: cumulative elevation change at the end of the natural and agricultural phase compared to the initial DEM for the different rainfall scenarios.**

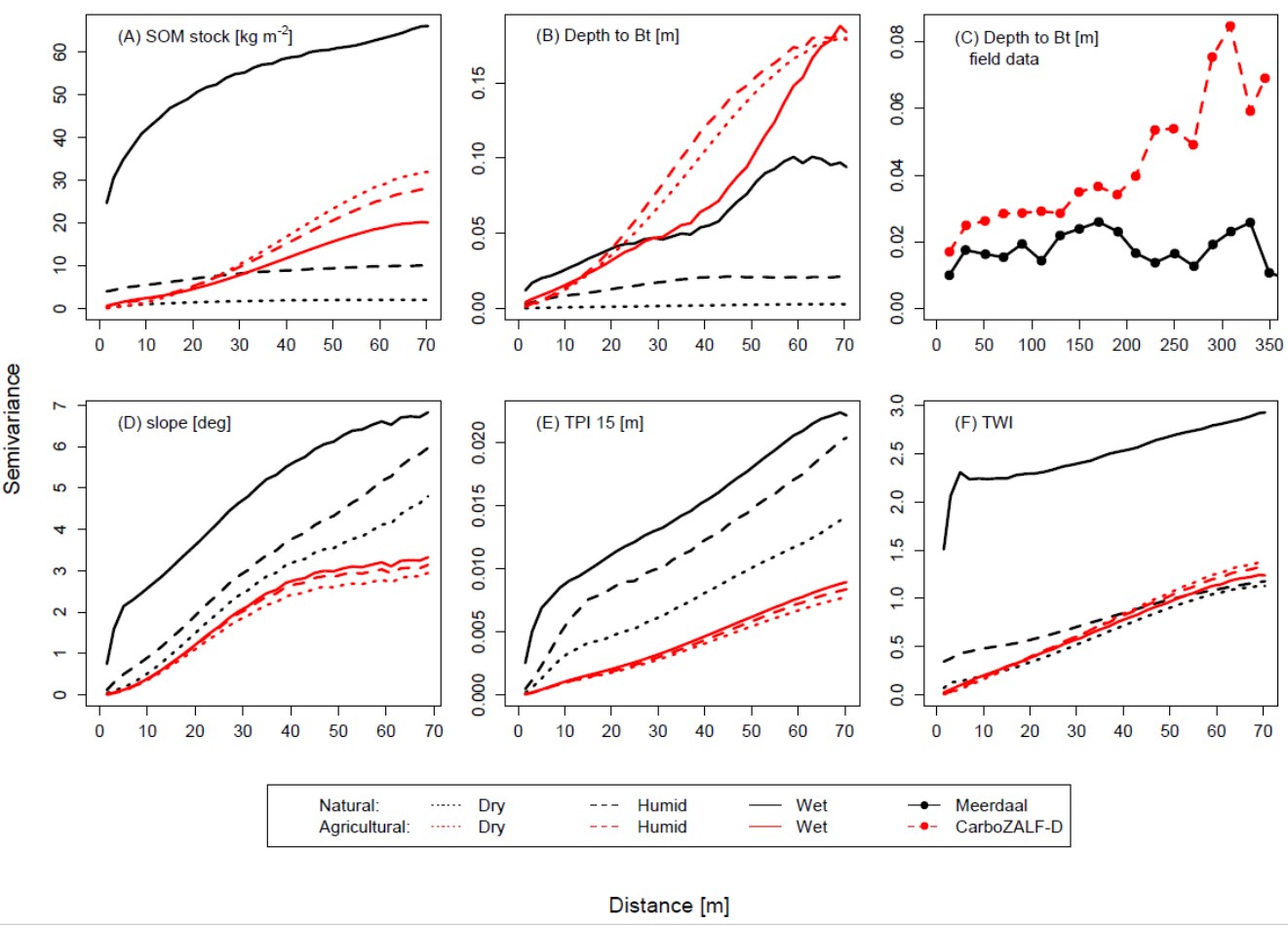

**Figure 6: Experimental semivariograms of the model results showing semivariance for different soil (A, B) and terrain properties (D-F) with different precipitation scenarios (line types) at the end of the natural (black) and agricultural (red) phases. For comparison, panel C shows experimental semivariograms of depth to Bt from a natural area (Meerdaal forest, P = 800 mm, Vanwalleghem et al., 2010) and an agricultural area (CarboZALF-D, P = 500 mm, Van der Meij et al., 2017). Note that these field data are presented with different axes. The experimental semivariograms are displayed with lines rather than points for easier visual comparison.**

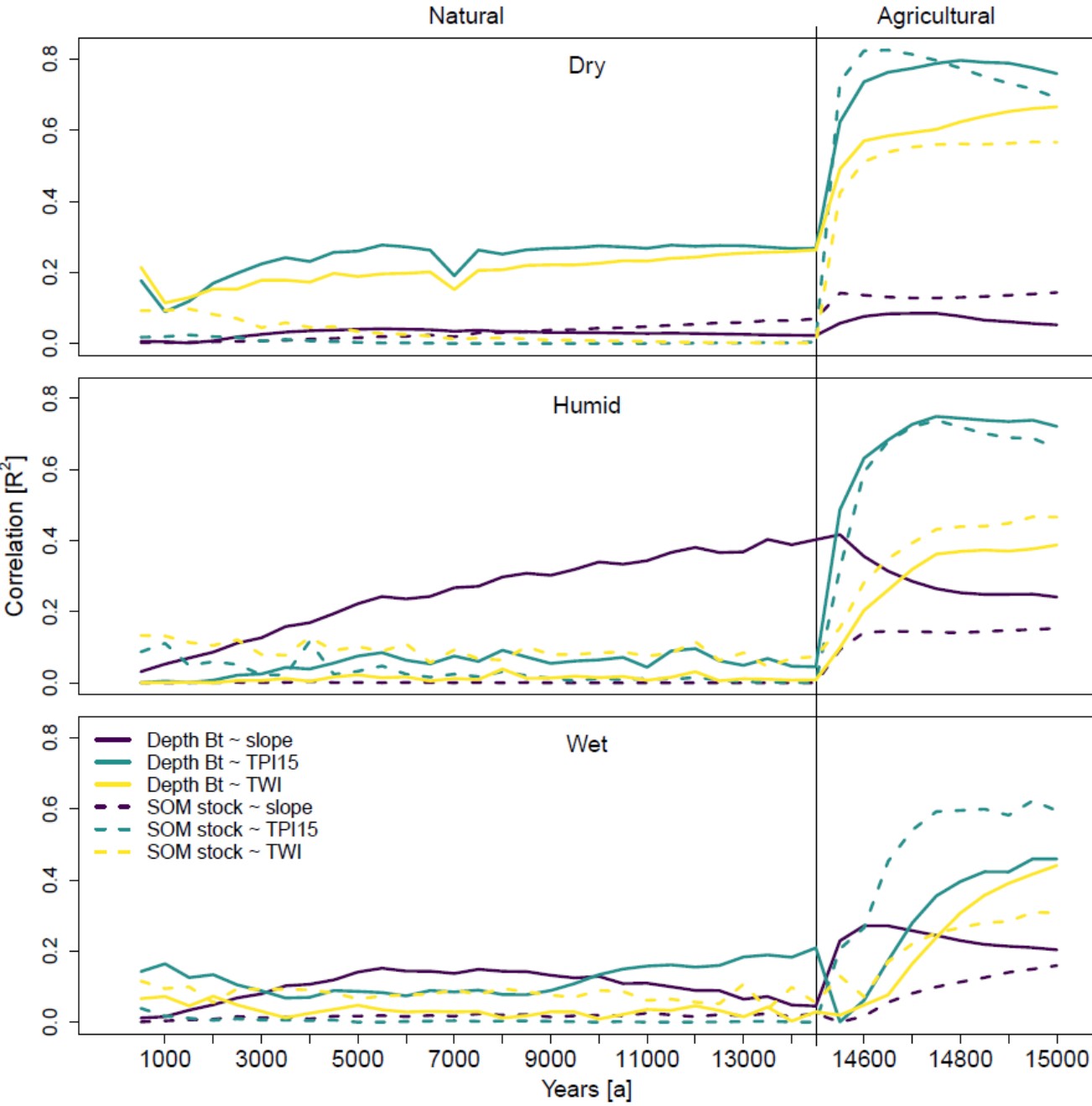

**Figure 7: Correlations (R2) between selected soil properties (line types) and topographic properties (colors) through time (left to right), for the different rainfall scenarios (top to bottom). In the natural system, the correlations are presented every 500 years, while in the agricultural system, the correlations are presented every 50 years. Note that for the latter phase the x-axis is stretched.**