# Peer review of "Modelling soil and landscape evolution– the effect of rainfall and land use change on soil and landscape patterns"

_SOIL, 2019_

## Referee Comment (RC1) · Anonymous Referee #1 · 19 Nov 2019

Review of "Modelling soil and landscape evolution- the effect of rainfall and land use change on soil and landscape patterns" authored by W. Marijn van der Meij, Arnaud J. A. M. Temme, Jakob Wallinga, Michael Sommer.

I found this a very interesting article that could prove to have a major impact. As far as I know it is the first time that water flow as driving pedogenetic process was added to a landscape evolution model, resulting in a (one of very few) non-empirical (but functional) soilscape model that could be used for global change studies. The article is well-written and a pleasure to read. My comments below focus on some points that would benefit from clarification or that could be named as assumptions behind some

choices in the modelling approach.

Remarks: 1. The authors have chosen to work with a hypothetical hilly landscape covered by loess rather than an existing landscape. This choice makes a full confrontation of model results to measurements impossible. This choice is understandable, as the history of real landscapes (and their agricultural history) is not easy to reconstruct and thus any inaccuracies could be resulting from the model, reconstructed boundary conditions, etc. Thus, a synthetic study is likely easier to interpret. The authors could pay some attention to this in their discussion: synthetic studies may avoid ambiguous interpretations of simulated versus real landscapes. At the same time, and as partial support, I would refer to http://dx.doi.org/10.1016/j.scitotenv.2016.07.119, where it was concluded that inaccuracies of boundary conditions over the simulation time did not significantly influence model results of the profile model SoilGen, while inaccuracies of initial conditions (like initial texture) did have significant influence.

2. The usage of bulk density estimated by a PTF like that by Tranter, to translate (simulated) mass per compartment to the volume (thickness) of that compartment makes good sense but is sensitive to the quality and the independent variables of that PTF. Tranter gives R2 of 0.49 of the best model, thus there is still considerable uncertainty. Furthermore, this PTF takes only texture and OM (as a proxy for soil structure) as inputs. In this sense, bulk density change (hence volume change) cannot be caused in the model by processes like decalcification and bioturbation. The authors smartly avoid the decalcification issue by assuming non-calcareous loess, which however limits the application domain. Bioturbation and tree throw are considered in the model but apparently do not directly affect bulk density. Perhaps these limitations could be mentioned in the discussion.

3. Unless I missed it, it seems to me that any climate change (in terms of precipitation) during the simulation period is absent (there are 3 scenarios, but these appear to be constant). It is well known that the precipitation surplus (as well as temperature) varied considerably, especially in the late glacial period but also afterwards. Can the authors

comment on possible effects that considering climate change might have had on co-evolution on soils and landscapes, additional to what has been stated already? I can imagine that cold and dry periods like Younger Dryas might have affected erosion for the reason that vegetation was less well developed or even absent. Is the reason not to include climate change related to the computational consequences of varying water flow dynamics?

4. I do not particularly like the 1:1 coupling of vegetation to infiltration regime; a forested site will not change into a grassland site on December 31st. There may be some more resilience there. This is also recognized by the authors, but I do not understand how they dealt with it. Lines 180-183 appear to suggest that outputs were time-aggregated, but inputs of vegetation type were not. Perhaps some clarification is useful. Btw, annual variation in infiltration is caused by the sum of precipitation and (re-)infiltration. Given the above remark, am I correct in concluding that the variation in re-infiltration is non-zero, whale the variation in P is zero? This would strengthen a terrain control on vegetation type, while there could also be a climate control. Additionally, for tree throws to result in a serious pit/mound topography, trees must have been present for a number of years and counting the "tree years" is not the strongest point in the model setting.

5. How thick is the loess, and what's below it? Line 195 states that shallow rooting depths do not occur (even after erosion), so the bottom of the loess is never reached? The effect of armouring (e.g. by coarse material originating from below the loess) on erosion is included in the model, so there seems to be no limitation there.

6. Can well-expressed Bt-horizons (such as present in Meerdaal as well) affect the rooting depth in the model?

7. line 433: SOM stocks in natural areas are estimated higher than often observed: Could this be because ectorganic material (O-horizon) is not simulated and thus this SOM is added to the mineral horizons?

8. line 574-577: I am not sure about the conclusion that in agricultural systems co-occurrence of non-interacting processes rather than co-evolution occurs. Reason: 14500 years of natural history are compared to 500 years of agricultural history. Is this a fair comparison? If you would compare the first 500 years of natural history to the 500 agriculture years, what would you conclude then?

---

## Referee Comment (RC2) · Christopher Shepard (Referee) · 31 Dec 2019

General Comments

The manuscript presented by van der Meij and others is interesting and well writ-ten, and will prove to be an important contribution to current pedogenic models. The manuscript presents a new formulation of the Lorica model, with the addition of a soil water balance component. The authors present model simulations for a 15 kyr period on a loess mantled landscape, with 14.5 kyr period of natural/ambient conditions and 500 yr period of intensive tilled agriculture; simulations were performed under three different rainfall scenarios.

[Figure]

I agree with the authors approach throughout much the manuscript. I think that reducing complexity, simplifying processes, and avoiding over parameterizations of models are principles that soil modelers should follow. There are several points that the authors could clarify or include throughout the manuscript.

Specific Comments -Intrinsic thresholds – There is a substantial discussion of progressive and regressive processes in the introduction, and a section about the potentially co-evolution/co-occurrence of soil-landscape processes and their relationship to external soil factors and drivers. The authors conclude that rainfall and landscape position, i.e. climate, are the dominant soil forming factor that generates soil spatial heterogeneity under ambient and agricultural contexts. However, there is no discussion of the possibility of intrinsic thresholds driving soil property change and variability across landscapes. Intrinsic thresholds also have the ability to create heterogeneity in soil properties without the influence of external soil factors. I think this may be an important component of soil evolution, particularly in natural settings, as some soil hydraulic changes can occur without changes in climate or water balance, such as argillic horizon formation leading to perched water tables and reducing conditions.

-Climate change – there was no discussion about the possible influence of climate change on pedogenesis throughout the manuscript. While this is a simulation, and as the authors note can only be used to understand general trends in pedogenesis, changes in climate over the last 15 kyr would like be a major driver of soil variability. However, this may be more of concern on a regional-continental scale, rather than the scale of the catchment consider in the present manuscript. Additionally, if we are to assume that these sites could exist at generally the same latitudes, I would expect soil responses to Holocene climate change at these hypothetical sites to be similar scaled.

-Vegetation switching – the authors tied vegetation type to the water availability. Depending on the annual water availability for the year, this may lead to annual transitions in vegetation. For this reason the authors consider vegetation type on multidecadal time scales. However, these "quick" transitions seem problematic in the model scheme.

Why were vegetation types not set for each simulation given that only one rainfall scenario could generate these transitions (humid)? Could the authors not have considered a savanna-type ecosystem for this precipitation level? I think this issue should be clarified in the manuscript text to aid in understanding of simulation parameterization and simulation results.

-Role of bulk density – based on the authors description it seems that estimating bulk density is central to estimating a number of soil variables from the model simulation, but there is little discussion in the main text on how this is done, other than with a PTF. I'm assuming this information is listed in the supplemental text. However, this should be included in the main text. Currently, it is unclear how these relate. This is especially important due the relationship between bulk density and OM and clay content. I may not understand the model architecture, but this would be greatly clarified with the inclusion of this information.

-Definition of intensive ag in loess mantled landscapes – The authors considered tilled agriculture in loess mantled landscapes. A recent trend, $\sim$50 years, of no-till ag has been prompted in many parts of North America, and I'm assuming the EU as well. Have the authors considered running similar simulations with no-till agriculture? This would be very timely, and may help us better understand SOM trends and long-term storage in soils in no till systems.

Technical Corrections

Line 89: "whereas" is not needed, please delete.

Line 90: "Therefore" is not needed, please delete.

Line 125: what are the two types of OM considered in the model?

Line 151: Please replace "didn't" with "did not".

Line 219: There are not other chemical information in the HydroLorica model. How did CEC evolve with the simulate soil landscape model?

[Figure]

Line 227: What is meant by SOM uptake? Accumulation in the soil? I would use a different choice of words for clarity.

Line 391-393: This sentence is unclear as written, I would remove the negative ("does not only"), and revise the sentence for clarity.

Line 408: Please delete "well" in "is well visible"; it is not needed

Line 410: Replace "get" with "become"

Line 436-438: I think that litter quality and input would also be a major driver of differences in SOM accumulation between natural and agricultural sites.

Line 450: "only" is not needed, please delete.

Line 476: "Especially" is not needed, please delete.

Line 511-512: Sentence starting "SOM cycling is heavily influenced..." This sentence is unclear as written. Please remove the "vice versa" and just say that erosion is not dependent on SOM cycling.

Line 515: I would also look at the work of Berhe et al. (2018). Ann. Rev. Earth and Plan. Sci. 46: 521 - 546, she has written extensively about the influence of erosion on SOM cycling.

Line 530-532: This sentence needs to be revised, it is not clear as written. Please revise the portion of the sentence starting with "...because changes in soil properties..."

---

## Author Comment (AC1) · 23 Jan 2020

Dear reviewer,

Thank you for your time and effort to review our paper. We appreciate your kind words and constructive remarks. Here we respond to your remarks one by one, with your comments in italic.

*I found this a very interesting article that could prove to have a major impact. As*

[Figure]

*far as I know it is the first time that water flow as driving pedogenetic process was added to a landscape evolution model, resulting in a (one of very few) non-empirical (but functional) soilscape model that could be used for global change studies. The article is well-written and a pleasure to read. My comments below focus on some points that would benefit from clarification or that could be named as assumptions behind some choices in the modelling approach*

*Remarks:*

*1. The authors have chosen to work with a hypothetical hilly landscape covered by loess rather than an existing landscape. This choice makes a full confrontation of model results to measurements impossible. This choice is understandable, as the history of real landscapes (and their agricultural history) is not easy to reconstruct and thus any inaccuracies could be resulting from the model, reconstructed boundary conditions, etc. Thus, a synthetic study is likely easier to interpret. The authors could pay some attention to this in their discussion: synthetic studies may avoid ambiguous interpretations of simulated versus real landscapes. At the same time, and as partial support, I would refer to http://dx.doi.org/10.1016/j.scitotenv.2016.07.119, where it was concluded that inaccuracies of boundary conditions over the simulation time did not significantly influence model results of the profile model SoilGen, while inaccuracies of initial conditions (like initial texture) did have significant influence.*

**Response:** The choice to simulate a synthetic landscape was indeed made to avoid a full confrontation of model results and field observations, because this comparison can be distorted by uncertainty in local climate history and land use history and potential other factors that might have played a role in the development of the current soil-landscape. Also, to compare the effect of different rainfall regimes on

soil-landscape evolution, as we did in the paper, other drivers of soil formation should preferably remain constant or vary only as consequence of the varying rainfall. We will expand our discussion on pros and cons of the simulation of synthetic landscapes in soil-landscape evolution in the revised version of the paper. The paper you suggested will be a useful support in this discussion.

*2. The usage of bulk density estimated by a PTF like that by Tranter, to translate (simulated) mass per compartment to the volume (thickness) of that compartment makes good sense but is sensitive to the quality and the independent variables of that PTF. Tranter gives R2 of 0.49 of the best model, thus there is still considerable uncertainty. Furthermore, this PTF takes only texture and OM (as a proxy for soil structure) as inputs. In this sense, bulk density change (hence volume change) cannot be caused in the model by processes like decalcification and bioturbation. The authors smartly avoid the decalcification issue by assuming non-calcareous loess, which however limits the application domain. Bioturbation and tree throw are considered in the model but apparently do not directly affect bulk density. Perhaps these limitations could be mentioned in the discussion.*

**Response:** The pedotransfer function (PTF) of Tranter et al. (2007) that we used required soil texture, SOM and depth below the surface as inputs. Depth below the surface acts as proxy for the effects of soil structure formation, weathering, bioturbation and other soil reworking. Tree throw and bioturbation do have an effect on the bulk density in the model through their effects on soil and layer thicknesses.

We agree that the uncertainty of this PTF is relatively high. However, PTFs that yield a higher accuracy often require soil hydrological or soil structural information,

which is not readily available in Lorica and HydroLorica. As we discuss in Van der Meij et al. (2018), the estimation of these parameters often gives biased or highly uncertain results, which would propagate into the calculation of bulk density. Rather than stacking pedotransfer functions, we decided to use a PTF that required input that is readily available in HydroLorica.

We will motivate the choice of our PTF in the Methods, describe the bulk density PTF in the manuscript and discuss the consequences of the chosen PTF on the model results in the Discussion to provide more information on this part of the model.

*3. Unless I missed it, it seems to me that any climate change (in terms of precipitation) during the simulation period is absent (there are 3 scenarios, but these appear to be constant). It is well known that the precipitation surplus (as well as temperature) varied considerably, especially in the late glacial period but also afterwards. Can the authors comment on possible effects that considering climate change might have had on coevolution on soils and landscapes, additional to what has been stated already? I can imagine that cold and dry periods like Younger Dryas might have affected erosion for the reason that vegetation was less well developed or even absent. Is the reason not to include climate change related to the computational consequences of varying water flow dynamics?*

**Response:** We indeed did not include the effects of a changing climate in our simulations. Next to a synthetic, simplified landscape, we also used a simplified climate scenario. The calculation demands would be a bit higher when more overland flow would occur, but this is not the reason we did not include climate dynamics. We agree that changes in climate have played a substantial role in the development of soils and

landscapes, especially more extreme climatic periods such as the Younger Dryas. However, the introduction of the model HydroLorica and simulation of three simplified rainfall scenarios already resulted in a complex and lengthy paper. Therefore, we think the effects of a changing climate on soil-landscape evolution are out of the scope of this paper, but this can be an interesting topic for a subsequent paper.

*4. I do not particularly like the 1:1 coupling of vegetation to infiltration regime; a forested site will not change into a grassland site on December 31st. There may be some more resilience there. This is also recognized by the authors, but I do not understand how they dealt with it. Lines 180-183 appear to suggest that outputs were time-aggregated, but inputs of vegetation type were not. Perhaps some clarification is useful. Btw, annual variation in infiltration is caused by the sum of precipitation and (re-)infiltration. Given the above remark, am I correct in concluding that the variation in re-infiltration is non-zero, whale the variation in P is zero? This would strengthen a terrain control on vegetation type, while there could also be a climate control. Additionally, for tree throws to result in a serious pit/mound topography, trees must have been present for a number of years and counting the "tree years" is not the strongest point in the model setting.*

**Response:** Vegetation type is indeed controlled by two factors: climate (precipitation) and terrain (re-infiltration). With these two factors we can for example simulate vegetation differentiation on north- and south-facing slopes and the occurrence of deciduous vegetation in locations where water flows converges, such as valleys and depressions (e.g. Metzen et al., 2019).

As we discuss in the paper and you indicate in your remark, we consider the

long-term effects of vegetation change on soil-landscape formation rather than the year-to-year variations (lines 180-183 in the original manuscript). This is similar to the simulation of clay translocations, where we consider the long-term changes in the soil profile rather than the differences between two consecutive years. We will clarify how we simulated and interpreted vegetation dynamics in the revised version of the manuscript.

The dimensions of the pit created by tree throw are a function of tree age (line 195-198 in the original manuscript, Eq. S6 in Supplement 1). This will lead to small root clumps for young trees, which will only cause a partial turbation of the upper layers in one raster cell in the simulations. Only when the dimensions of the root clump exceed the size of a raster cell (radius of 1.5 m in our case), a pit-and-mount topography is created. We hope that this explanation resolves your concerns on the creation of pit-and-mound topography in our simulations. We will clarify this point in the revised manuscript.

*5. How thick is the loess, and what's below it? Line 195 states that shallow rooting depths do not occur (even after erosion), so the bottom of the loess is never reached? The effect of armouring (e.g. by coarse material originating from below the loess) on erosion is included in the model, so there seems to be no limitation there.*

**Response:** In our simulations we assumed an infinite layer of loess to avoid potential effects of lower layers. However, for computational reasons we worked with an initial loess layer of three meters with free leaching of water and potential clay at the bottom of the soil columns. This approach reduced the amount of soil layers, avoided numerical instability from the pedotransfer function for bulk density which is

depth-dependent and was still thick enough that the bottom of the loess was never reached by erosion. We forgot to mention loess thickness and the free leaching in the original version of the manuscript and we will include this in the revised version.

The shallow rooting depths we refer to in line 195 can be a cause of tree throw, for example when the roots are blocked by rocks or impermeable soil layers. We did not include such limitations in our simulations, but this would likely not have occurred since we limited the thickness of the root clump for tree throw to a maximum of 70 cm. For the calculation of bioturbation and SOM cycling, we varied the potential rates to account for rooting differences between vegetations. The effect of armouring is indeed included in HydroLorica, but did not play a role in our simulations, because there was no coarse fraction present that would lead to armouring.

*6. Can well-expressed Bt-horizons (such as present in Meerdaal as well) affect the rooting depth in the model?*

**Response:** As we stated at the previous remark, we did not include such limitations in our simulations. Depending on the settings, Bt horizons can limit root growth, but also facilitate root growth by structure formation and increased nutrient availability. The occurrence, type and grade of soil structuring is very difficult to estimate and therefore we did not consider this effect in this paper.

*7. line 433: SOM stocks in natural areas are estimated higher than often observed: Could this be because ectorganic material (O-horizon) is not simulated and thus this SOM is added to the mineral horizons?*

**Response:** We indeed did not include the formation of O-horizons in our simulations. However, we compared the stocks from our simulations with soils including O-horizons from the paper of Wiesmeier et al. (2012), and still our estimations were higher. As we argue in the paper, the agriculturally-derived SOM depth profiles were not representative for forested sites, because other factors and processes affected uptake and decay of soil organic matter under forested conditions (lines 436-439). We are currently not able to simulate and calibrate these processes properly. We will mention this in the revised manuscript.

*8. line 574-577: I am not sure about the conclusion that in agricultural systems cooccurrence of non-interacting processes rather than co-evolution occurs. Reason: 14500 years of natural history are compared to 500 years of agricultural history. Is this a fair comparison? If you would compare the first 500 years of natural history to the 500 agriculture years, what would you conclude then?*

**Response:** We derived this conclusion from our findings that under natural conditions the formation of soils, terrain, the hydrological system and vegetation are intertwined. Changes in one domain in the landscape have effects on the formation of all other domains. These interactions, or co-evolution, occur on both short and long timescales, but become more pronounced over longer timescales, due to progressive soil and landscape formation. This is visible in the animation in Supplement 2, where there are already considerable differences between the soil patterns from each scenario after 500 years of natural soil formation, due to the role of water and vegetation in soil-landscape co-evolution. These differences become more pronounced over time. In comparison, the differences between the patterns after 500 years of agricultural

land use are much smaller. Anthropogenic processes do not show co-evolution, because the rates of for example tillage erosion far exceed any rates of natural soil and landscape change (see Fig. 5 in the manuscript). Tillage can introduce new processes or accelerate other processes e.g. by breaking up aggregates. However, these processes do not affect the rate at which a plough transports sediments through a landscape. If these interactions do not occur on shorter timescales, they will also not emerge over longer timescales. We think that the differences in time scales between the two land use periods do not affect our conclusion, because co-evolution is not time-dependent, but process-dependent.

**References**

Metzen, D., Sheridan, G.J., Benyon, R.G., Bolstad, P.V., Griebel, A., Lane, P.N.J. 2019. Spatio-temporal transpiration patterns reflect vegetation structure in complex upland terrain. Science of The Total Environment 694: 133551. https://doi.org/10.1016/j.scitotenv.2019.07.357

Tranter, G., Minasny, B., Mcbratney, A.B., Murphy, B., Mckenzie, N.J., Grundy, M., Brough, D. 2007. Building and testing conceptual and empirical models for predicting soil bulk density. Soil Use and Management 23 (4): 437-443. https://doi.org/10.1111/j.1475-2743.2007.00092.x

Van der Meij, W.M., Temme, A.J.A.M., Lin, H.S., Gerke, H.H., Sommer, M. 2018. On the role of hydrologic processes in soil and landscape evolution modeling: concepts, complications and partial solutions. Earth-Science Reviews 185: 1088-1106. https://doi.org/10.1016/j.earscirev.2018.09.001

Wiesmeier, M., Spörlein, P., Geuß, U., Hangen, E., Haug, S., Reischl, A., Schilling, B., von Lützow, M., Kögel‐Knabner, I. 2012. Soil organic carbon stocks in southeast Germany (Bavaria) as affected by land use, soil type and sampling depth. Global Change Biology 18 (7): 2233-2245. https://doi.org/10.1111/j.1365-2486.2012.02699.x

---

## Author Comment (AC2) · 23 Jan 2020

Dear Dr. Shepard,

Thank you for your extensive review of our manuscript and the encouraging words. We reply to your comments below one-by-one, with your comments in italic.

*General Comments*
*The manuscript presented by van der Meij and others is interesting and well written,*

[Figure]

*and will prove to be an important contribution to current pedogenic models. The manuscript presents a new formulation of the Lorica model, with the addition of a soil water balance component. The authors present model simulations for a 15 kyr period on a loess mantled landscape, with 14.5 kyr period of natural/ambient conditions and 500 yr period of intensive tilled agriculture; simulations were performed under three different rainfall scenarios.*

*I agree with the authors approach throughout much the manuscript. I think that reducing complexity, simplifying processes, and avoiding over parameterizations of models are principles that soil modelers should follow. There are several points that the authors could clarify or include throughout the manuscript.*

*Specific Comments*
*-Intrinsic thresholds – There is a substantial discussion of progressive and regressive processes in the introduction, and a section about the potentially co-evolution/co-occurrence of soil-landscape processes and their relationship to external soil factors and drivers. The authors conclude that rainfall and landscape position, i.e. climate, are the dominant soil forming factor that generates soil spatial heterogeneity under ambient and agricultural contexts. However, there is no discussion of the possibility of intrinsic thresholds driving soil property change and variability across landscapes. Intrinsic thresholds also have the ability to create heterogeneity in soil properties without the influence of external soil factors. I think this may be an important component of soil evolution, particularly in natural settings, as some soil hydraulic changes can occur without changes in climate or water balance, such as argillic horizon formation leading to perched water tables and reducing conditions.*

**Response:** We acknowledge the role that intrinsic thresholds can play in the evolution of soils and landscapes. We discuss these thresholds and following

co-evolution in van der Meij et al. (2018). But, as we mention in the same paper, such intrinsic thresholds can currently not be modelled, because we lack the methods for estimating accurate soil hydraulic properties which drive the threshold behavior.

Ideally, the model shows such threshold behavior without explicitly incorporating these thresholds in the model code. However, such hard thresholds can cause problems when calibrating the model by creating sharp discontinuities in the model results as a response to slight variations in parameters (Barnhart et al., 2019). Also, the occurrence of such thresholds depends on a large variety of factors, such as clay type and clay content, moisture dynamics and land management. When such factors cannot all be modeled and a hard threshold is assumed based on one of these factors (e.g. clay content), the model can give wrong results. For these reasons we focus on heterogeneity related to external causes in this manuscript. We will mention the potential role of intrinsic thresholds on soil heterogeneity in the revised manuscript and we will argue why we did not include them in our simulations.

*-Climate change – there was no discussion about the possible influence of climate change on pedogenesis throughout the manuscript. While this is a simulation, and as the authors note can only be used to understand general trends in pedogenesis, changes in climate over the last 15 kyr would like be a major driver of soil variability. However, this may be more of concern on a regional-continental scale, rather than the scale of the catchment consider in the present manuscript. Additionally, if we are to assume that these sites could exist at generally the same latitudes, I would expect soil responses to Holocene climate change at these hypothetical sites to be similar scaled.*

**Response:** We agree that changes in climate have played a major role in soil and landscape variability over the Holocene. However, our aim was to isolate the effects of different rainfall and land use regimes on soil and landscape evolution. We decided to vary only the amount of rainfall to reduce the amount of variables that could have influenced the model results. The effects of a changing climate on soil-landscape evolution are thus out of the scope of this paper, but this can be an interesting topic for a subsequent paper. We will mention in the revised manuscript that we simulated a present-day Holocene climate, but that our simulation period extends beyond this climate period.

*-Vegetation switching – the authors tied vegetation type to the water availability. Depending on the annual water availability for the year, this may lead to annual transitions in vegetation. For this reason the authors consider vegetation type on multidecadal time scales. However, these "quick" transitions seem problematic in the model scheme. Why were vegetation types not set for each simulation given that only one rainfall scenario could generate these transitions (humid)? Could the authors not have considered a savanna-type ecosystem for this precipitation level? I think this issue should be clarified in the manuscript text to aid in understanding of simulation parameterization and simulation results.*

**Response:** We linked vegetation type to moisture availability to include both the effects of hillslope aspect and local convergence of water flow to gulleys or depressions on vegetation type (e.g. Metzen et al., 2019). This variation in moisture and vegetation can occur very locally, especially in semi-arid regions. On top of this spatial variation, the infiltration patterns also vary in time, due to changing infiltration capacity and surface water routing. Because of this co-evolution of soils, terrain,

vegetation and the hydrological system, we think it is not a good idea to fix vegetation type depending on external rainfall amount. We will clarify the choice for dynamic vegetation modeling in the manuscript.

*-Role of bulk density – based on the authors description it seems that estimating bulk density is central to estimating a number of soil variables from the model simulation, but there is little discussion in the main text on how this is done, other than with a PTF. I'm assuming this information is listed in the supplemental text. However, this should be included in the main text. Currently, it is unclear how these relate. This is especially important due the relationship between bulk density and OM and clay content. I may not understand the model architecture, but this would be greatly clarified with the inclusion of this information.*

**Response:** The PTF for bulk density is indeed not properly discussed in the reviewed manuscript. We will include the description of the used pedotransfer function in the Methodology section of the manuscript and we will discuss the consequences of the chosen PTF on the model output in the Discussion of the revised manuscript.

*-Definition of intensive ag in loess mantled landscapes – The authors considered tilled agriculture in loess mantled landscapes. A recent trend, ~50 years, of no-till ag has been prompted in many parts of North America, and I'm assuming the EU as well. Have the authors considered running similar simulations with no-till agriculture? This would be very timely, and may help us better understand SOM trends and long-term storage in soils in no till systems.*

**Response:** Reduced-tillage or no-tillage is indeed also an occurring trend in Europe to reduce land degradation. But, just as with the rainfall, we simulated a simple scenario of intensive agriculture to facilitate interpretation of the model results. The implications of different tillage regimes on soil-landscape evolution can be the topic of a subsequent study.

*Technical Corrections*

**Response:** We will address the questions and process the corrections in the revised manuscript.

*Line 89: "whereas" is not needed, please delete.*
*Line 90: "Therefore" is not needed, please delete.*
*Line 125: what are the two types of OM considered in the model?*
*Line 151: Please replace "didn't" with "did not".*
*Line 219: There are not other chemical information in the HydroLorica model. How did CEC evolve with the simulate soil landscape model?*
*Line 227: What is meant by SOM uptake? Accumulation in the soil? I would use a different choice of words for clarity.*
*Line 391-393: This sentence is unclear as written, I would remove the negative ("does not only"), and revise the sentence for clarity.*
*Line 408: Please delete "well" in "is well visible"; it is not needed*
*Line 410: Replace "get" with "become"*
*Line 436-438: I think that litter quality and input would also be a major driver of differences in SOM accumulation between natural and agricultural sites.*
*Line 450: "only" is not needed, please delete. Line 476: "Especially" is not needed, please delete.*

*Line 511-512: Sentence starting "SOM cycling is heavily influenced. . ." This sentence is unclear as written. Please remove the "vice versa" and just say that erosion is not dependent on SOM cycling.*
*Line 515: I would also look at the work of Berhe et al. (2018). Ann. Rev. Earth and Plan. Sci. 46: 521 - 546, she has written extensively about the influence of erosion on SOM cycling.*
*Line 530-532: This sentence needs to be revised, it is not clear as written. Please revise the portion of the sentence starting with ". . .because changes in soil properties. . ."*

**References**

Barnhart, K.R., Glade, R.C., Shobe, C.M., Tucker, G.E. 2019. Terrainbento 1.0: a Python package for multi-model analysis in long-term drainage basin evolution. Geoscientific Model Development 12 (4): 1267-1297. https://doi.org/10.5194/gmd-12-1267-2019

Metzen, D., Sheridan, G.J., Benyon, R.G., Bolstad, P.V., Griebel, A., Lane, P.N.J. 2019. Spatio-temporal transpiration patterns reflect vegetation structure in complex upland terrain. Science of The Total Environment 694: 133551. https://doi.org/10.1016/j.scitotenv.2019.07.357

van der Meij, W.M., Temme, A.J.A.M., Lin, H.S., Gerke, H.H., Sommer, M. 2018. On the role of hydrologic processes in soil and landscape evolution modeling: concepts, complications and partial solutions. Earth-Science Reviews 185: 1088-1106. https://doi.org/10.1016/j.earscirev.2018.09.001

---

## Author Response (AR2)

Dear professor Fiener,

Thank you for your comments on our manuscript. We adopted most of your suggestions in our manuscript. Below we mention the changes and motivate why we did or did not adopt some of your suggestions. The page and line numbers refer to the manuscript with marked changes. We hope that, with these modifications, you deem the paper fit for publication.

On behalf of all authors, with best regards,

Marijn van der Meij

**Geostatistical terminology**

We changed the terminology of the geostatistics and used the proper terms to describe the scale and magnitude of soil and landscape variability (lines 387-414). We removed the comparison about the shape of the semivariograms from model and field results (lines 452-466). Instead, we only describe the differences between a natural and an agricultural setting and we mention why the magnitudes of the range, sill and nugget are different between model and field results. Although the experimental semivariograms in Fig. 6 should indeed be represented by points as you indicate, we think that using lines gives a calmer image which allows easier visual comparison. Also, the line types represent the different gradations of rainfall, which is not possible with points. That is why we decided to use the lines instead. We motivate this choice in the caption of Fig. 6 (line 1060).

**Applicability of our findings for different agricultural fields**

We agree with your remarks about the limited applicability of our findings and conclusions for all types of agricultural fields. Although we expect that the identified soil-landscape relations will occur in every tilled field, the variation within a field might be secondary to variation emerging from other factors such as extensive management and field size, land use history and field boundaries such as banks or hedges, as you suggest. We added these nuances throughout our manuscript (lines 54-55, 109-110, 446-451, 484-488, 549-550, 666), and limited our hypotheses and conclusions to large-field settings which are representative for intensive agricultural use (lines 105-110, 689-691, 700).

**Model description**

- The simulation of a coarse fraction is a functionality in HydroLorica which is available, but which we did not use in our simulations. We added this notion to the manuscript (lines 129, 143-146).
- We use the brief discussion of the quality of the used PTF (lines 146-153) to motivate our choice to use this PTF. Therefore, we think this part is in the right place in the Methodology section.

**Minor comments**

We adopted your additional suggestions to clarify the manuscript.

[revised manuscript text omitted]